# Molecular dissection of PI3Kβ synergistic activation by receptor tyrosine kinases, GβGγ, and Rho-family GTPases

**Benjamin R Duewell[†], Naomi E Wilson[†], Gabriela M Bailey, Sarah E Peabody, Scott D Hansen***

Department of Chemistry and Biochemistry, Institute of Molecular Biology, University of Oregon, Eugene, United States

**\*For correspondence:**
shansen5@uoregon.edu

[†]These authors contributed equally to this work

**Competing interest:** The authors declare that no competing interests exist.

**Abstract** Phosphoinositide 3-kinase (PI3K) beta (PI3Kβ) is functionally unique in the ability to integrate signals derived from receptor tyrosine kinases (RTKs), G-protein coupled receptors, and Rho-family GTPases. The mechanism by which PI3Kβ prioritizes interactions with various membrane-tethered signaling inputs, however, remains unclear. Previous experiments did not determine whether interactions with membrane-tethered proteins primarily control PI3Kβ localization versus directly modulate lipid kinase activity. To address this gap in our knowledge, we established an assay to directly visualize how three distinct protein interactions regulate PI3Kβ when presented to the kinase in a biologically relevant configuration on supported lipid bilayers. Using single molecule Total Internal Reflection Fluorescence (TIRF) Microscopy, we determined the mechanism controlling PI3Kβ membrane localization, prioritization of signaling inputs, and lipid kinase activation. We find that auto-inhibited PI3Kβ prioritizes interactions with RTK-derived tyrosine phosphorylated (pY) peptides before engaging either GβGγ or Rac1(GTP). Although pY peptides strongly localize PI3Kβ to membranes, stimulation of lipid kinase activity is modest. In the presence of either pY/GβGγ or pY/Rac1(GTP), PI3Kβ activity is dramatically enhanced beyond what can be explained by simply increasing membrane localization. Instead, PI3Kβ is synergistically activated by pY/GβGγ and pY/Rac1 (GTP) through a mechanism consistent with allosteric regulation.

## eLife assessment

The manuscript describes the synergy among PI3Kβ activators, providing **compelling** results concerning the mechanism of their activation. The particular strengths of the work arise to a great extend from the reconstitution system better mimicking the natural environment of the plasma membrane than previous setups have. The study will be a **landmark** contribution to the signaling field.

## Introduction

Critical for cellular organization, phosphatidylinositol phosphate (PIP) lipids regulate the localization and activity of numerous proteins across intracellular membranes in eukaryotic cells (*Di Paolo and De Camilli, 2006*). The interconversion between various PIP lipid species through the phosphorylation and dephosphorylation of inositol head groups is regulated by lipid kinases and phosphatases (*Balla, 2013*; *Burke, 2018*). Serving a critical role in cell signaling, the class I family of PI3Ks catalyze the phosphorylation of phosphatidylinositol 4,5-bisphosphate [PI(4,5)P$_2$] to generate PI(3,4,5)P$_3$. Although a low-abundance lipid (<0.05%) in the plasma membrane (*Wenk et al., 2003*; *Nasuhoglu et al., 2002*; *Stephens et al., 1993*), PI(3,4,5)P$_3$ can increase 40-fold following receptor

activation (*Stephens et al., 1991*; *Parent et al., 1998*; *Insall and Weiner, 2001*). Although signal adaptation mechanisms typically restore PI(3,4,5)P$_3$ to the basal level following receptor activation (*Funamoto et al., 2002*; *Yip et al., 2008*; *Auger et al., 1989*), misregulation of the PI3K signaling pathway can result in constitutively high levels of PI(3,4,5)P$_3$ that are detrimental to cell health. Since PI(3,4,5)P$_3$lipids serve an instructive role in driving actin-based membrane protrusions (*Howard and Oresajo, 1985*; *Weiner, 2002*; *Graziano et al., 2017*), sustained PI(3,4,5)P$_3$ signaling is known to drive cancer cell metastasis (*Hanker et al., 2013*). Elevated PI(3,4,5)P$_3$ levels also stimulates the AKT signaling pathway and Tec family kinases, which can drive cellular proliferation and tumorigenesis (*Manning and Cantley, 2007*; *Fruman et al., 2017*). While much work has been dedicated in determining the factors that participate in the PI3K signaling pathway, how these molecules collaborate to rapidly synthesize PI(3,4,5)P$_3$ remains an important open question. To decipher how amplification of PI(3,4,5)P$_3$ arises from the relay of signals between cell surface receptors, lipids, and peripheral membrane proteins, we must understand how membrane localization and activity of PI3Ks is regulated by different signaling inputs. This will provide new insight concerning the molecular basis of asymmetric cell division, cell migration, and tissue organization, which are critical for understanding development and tumorigenesis.

In the absence of a stimulatory input, the class IA family of PI3Ks (PI3Kα, PI3Kβ, PI3Kδ) are thought to reside in the cytoplasm as auto-inhibited heterodimeric protein complexes composed of a catalytic (p110α, p110β, or p110δ) and regulatory subunit (p85α, p85β, p55γ, p50α, or p55α) (*Burke, 2018*; *Vadas et al., 2011*). The catalytic subunits of class IA PI3Ks contain an N-terminal adaptor binding domain (ABD), a Ras/Rho binding domain (RBD), a C2 domain (C2), and an adenosine triphosphate (ATP) binding pocket (*Vadas et al., 2011*). The inter-SH2 (iSH2) domain of the regulatory subunit tightly associates with the ABD of the catalytic subunit (*Yu et al., 1998*), providing structural integrity, while limiting dynamic conformational changes. The nSH2 and cSH2 domains of the regulatory subunit form additional inhibitory contacts that limit the conformational dynamics of the catalytic subunit (*Zhang et al., 2011*; *Mandelker et al., 2009*; *Burke et al., 2011*; *Carpenter et al., 1993*; *Yu et al., 1998*). A clearer understanding of how various proteins control PI3K localization and activity would help facilitate the development of drugs that perturb specific protein-protein binding interfaces critical for membrane targeting and lipid kinase activity.

Among the class IA PI3Ks, PI3Kβ is uniquely capable of interacting with Rho-family GTPases (*Fritsch et al., 2013*), Rab GTPases (*Christoforidis et al., 1999*; *Heitz et al., 2019*), heterotrimeric G-protein complexes (GβGγ) (*Kurosu et al., 1997*; *Maier et al., 1999*; *Guillermet-Guibert et al., 2008*), and phosphorylated receptor tyrosine kinases (RTKs) (*Zhang et al., 2011*; *Carpenter et al., 1993*). Like other class IA PI3Ks, interactions with RTK derived phosphotyrosine peptides release nSH2 and cSH2-mediated inhibition of the catalytic subunit to stimulate PI3Kβ lipid kinase activity (*Dbouk et al., 2012*; *Zhang et al., 2011*). GβGγ and Rac1(GTP) in solution have also been shown to stimulate PI3Kβ lipid kinase activity (*Dbouk et al., 2012*; *Fritsch et al., 2013*; *Maier et al., 1999*). Similarly, activation of Rho-family GTPases (*Fritsch et al., 2013*) and G-protein coupled receptors (*Houslay et al., 2016*) in cells, stimulates PI3Kβ lipid kinase activity. However, it's unclear how individual interactions with GβGγ or Rac1(GTP) can bypass autoinhibition of full-length PI3Kβ (p110β-p85α/β). Studies in neutrophils and in vitro biochemistry suggest that PI3Kβ is synergistically activated through coincidence detection of RTKs and GβGγ (*Houslay et al., 2016*; *Dbouk et al., 2012*). Similarly, Rac1(GTP) and GβGγ have been reported to synergistically activate PI3Kβ in cells (*Erami et al., 2017*). An enhanced membrane recruitment mechanism is the most prominent model used to explain synergistic activation of PI3Ks.

There is limited kinetic data examining how PI3Kβ is regulated by different membrane-tethered proteins. Previous biochemical studies of PI3Kβ have utilized solution-based assays to measure P(3,4,5)P$_3$ production. As a result, the mechanisms that determine how PI3Kβ prioritizes interactions with RTKs, small GTPases, or GβGγ remains unclear. In the case of synergistic PI3Kβ activation, it's unclear which protein-protein interactions regulate membrane localization versus stimulate lipid kinase activity. No studies have simultaneously measured PI3Kβ membrane association and lipid kinase activity to decipher potential mechanisms of allosteric regulation. Previous studies concerning the synergistic activation of PI3Ks are challenging to interpret because RTK-derived peptides are always presented in solution alongside membrane-anchored signaling inputs. However, all the common signaling inputs for PI3K activation (i.e. RTKs, GβGγ, Rac1/Cdc42) are membrane-associated proteins. Activation of class 1A PI3Ks has never been reconstituted using solely membrane-tethered activators conjugated

to membranes in a biologically relevant configuration. As a result, we currently lack a comprehensive description of PI3Kβ membrane recruitment and catalysis.

To decipher the mechanisms controlling PI3Kβ membrane binding and activation, we established a biochemical reconstitution using supported lipid bilayers (SLBs). We used single molecule Total Internal Reflection Fluorescence (TIRF) microscopy to quantify the relationship between PI3Kβ localization, lipid kinase activity, and the density of various membrane-tethered signaling inputs. This approach allowed us to measure the dwell time, binding frequency, and diffusion coefficients of single fluorescently labeled PI3Kβ in the presence of RTK-derived peptides, Rac1(GTP), and GβGγ. Simultaneous measurements of PI3Kβ membrane recruitment and lipid kinase activity allowed us to define the relationship between PI3Kβ localization and PI(3,4,5)P$_3$ production in the presence of different regulators. Overall, we found that membrane docking of PI3Kβ first requires interactions with RTK-derived tyrosine phosphorylated (pY) peptides, while PI3Kβ localization is insensitive to membranes that contain either Rac1(GTP) or GβGγ alone. Following engagement with a pY peptide, PI3Kβ can associate with either GβGγ or Rac1(GTP). In the case of synergistic PI3Kβ localization mediated by pY/GβGγ, it's essential for the nSH2 domain to move away from the GβGγ binding site. Although both the PI3Kβ-pY-Rac1(GTP) and PI3Kβ-pY-GβGγ complexes display a ~2-fold increase in membrane localization, the corresponding increase in catalytic efficiency is much greater. Overall, our results indicate that synergistic activation of PI3Kβ depends on allosteric modulation of lipid kinase activity.

## Results

### PI3Kβ prioritizes interactions with pY peptides over Rac1(GTP) and GβGγ

Previous biochemical analysis of p110β-p85α, referred to as PI3Kβ, established that receptor tyrosine kinases (*Zhang et al., 2011*), Rho-type GTPases (*Fritsch et al., 2013*), and heterotrimeric G-protein GβGγ (*Dbouk et al., 2012*) are capable of binding and stimulating lipid kinase activity. To decipher how PI3Kβ prioritizes interactions between these three membrane-tethered proteins we established a method to directly visualize PI3Kβ localization on supported lipid bilayers (SLBs) using Total Internal Reflection Fluorescence (TIRF) Microscopy (*Figure 1A*). For this assay, we covalently attached either a doubly tyrosine phosphorylated platelet-derived growth factor (PDGF) peptide (pY) peptide or recombinantly purified Rac1 to supported membranes using cysteine reactive maleimide lipids. We confirmed membrane conjugation of the pY peptide and Rac1 by visualizing the localization of fluorescently labeled nSH2-Cy5 or Cy3-p67/phox (Rac1(GTP) sensor), respectively (*Figure 1B*). Nucleotide exchange of membrane conjugated Rac1(GDP) was achieved by the addition of a guanine nucleotide exchange factor, P-Rex1 (phosphatidylinositol 3,4,5-trisphosphate-dependent Rac exchanger 1 protein) diluted in GTP containing buffer (*Figure 1C*). As previously described (*Rathinaswamy et al., 2021*; *Rathinaswamy et al., 2023*), AF488-SNAP dye-labeled farnesyl GβGγ was directly visualized following passive absorption into supported membranes (*Figure 1D* and *Figure 1—figure supplement 1*). We confirmed that membrane-bound GβGγ was functional by visualizing robust membrane recruitment of Dy647-PI3Kγ by TIRF-M (*Figure 1E*). Overall, this assay functions as a mimetic to the cellular plasma membrane and allowed us to examine how different membrane-tethered signaling inputs regulate PI3Kβ membrane localization in vitro.

We visualized both single molecule binding events and bulk membrane localization of Dy647-PI3Kβ by TIRF microscopy to determine which inputs can autonomously recruit autoinhibited Dy647-PI3Kβ from solution to a supported membrane (*Figure 1F*). Comparing membrane localization of Dy647-PI3Kβ in the presence of pY, Rac1(GTP), or GβGγ revealed that only the tyrosine phosphorylated peptide (pY) could robustly localize Dy647-PI3Kβ to supported membranes (*Figure 1F–G*). This prioritization of interactions was consistently observed across a variety of membrane lipid compositions (*Figure 1—figure supplement 2*). Increasing the anionic lipid charge of supported membranes through the addition of 20% phosphatidylserine (PS), did not significantly change the frequency of Dy647-PI3Kβ membrane interactions in the presence of either Rac1(GTP) or GβGγ alone (*Figure 1—figure supplement 2*). Although we could detect some transient Dy647-PI3Kβ membrane binding events in the presence of GβGγ alone, the binding frequency was reduced 2000-fold compared to our measurements on pY membranes (*Figure 1—figure supplement 2*). Localization of wild-type Dy647-PI3Kβ phenocopied the GβGγ binding mutant, Dy647-PI3Kβ (K532D, K533D)**,** indicating that

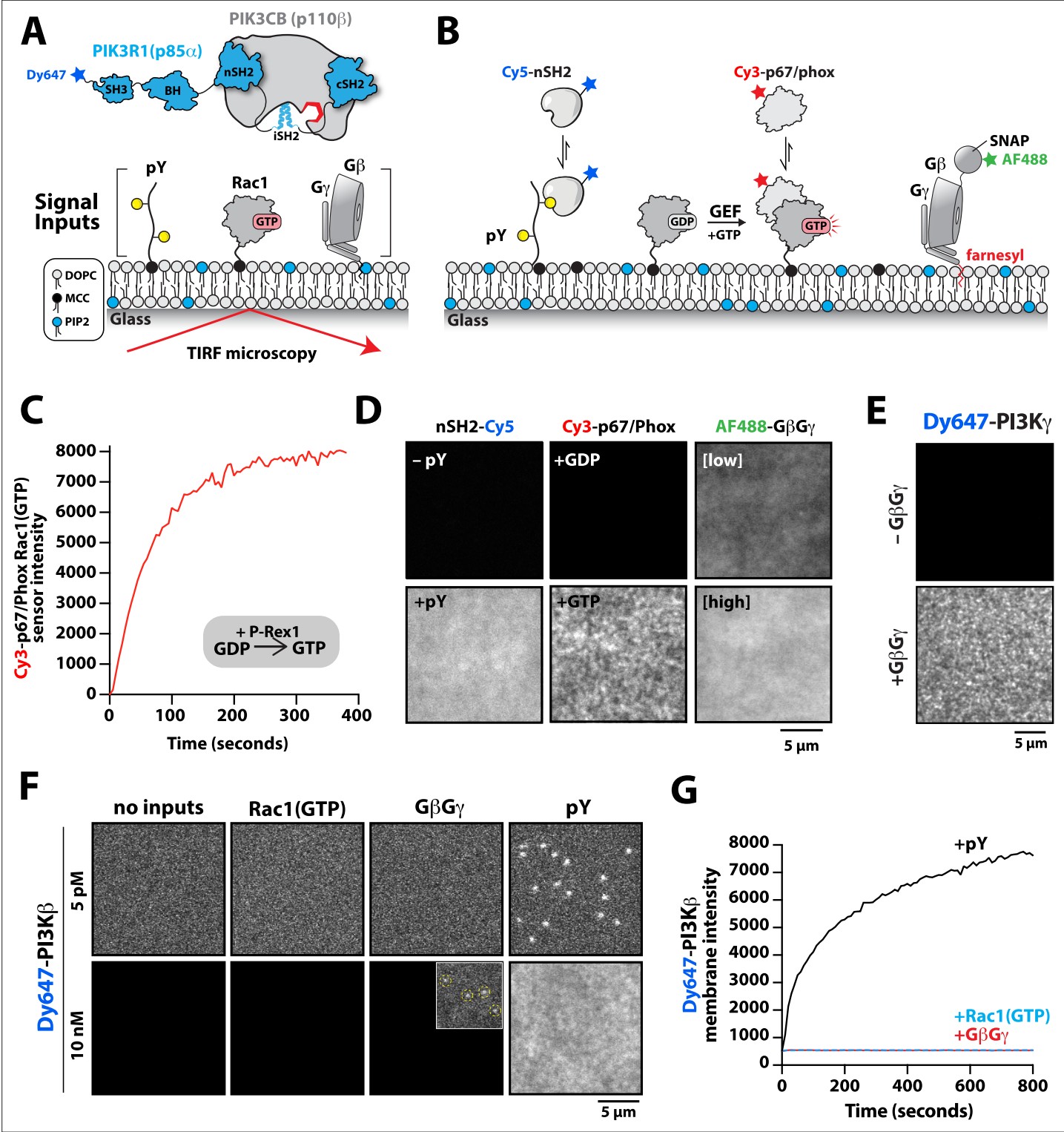

**Figure 1.** Phosphoinositide 3-kinase beta (PI3Kβ) prioritizes membrane interactions with receptor tyrosine kinase (RTK)-derived phosphorylated (pY) peptides over Rac1(GTP) and G-protein complexes (GβGγ). (**A**) Cartoon schematic showing membrane-tethered signaling inputs (i.e. pY, Rac1(GTP), and GβGγ) attached to a supported lipid bilayer and visualized by TIRF-M. Heterodimeric Dy647-PI3Kβ (p110β-p85α) in solution can dynamically associate with membrane-bound proteins. (**B**) Cartoon schematic showing method for visualizing membrane-tethered signaling inputs. (**C**) Kinetics of Rac1 nucleotide exchange measured in the presence of 20 nM Rac1(GTP) sensor (Cy3-p67/phox) and 50 nM P-Rex1 (DH-PH domain). (**D**) Visualization of membrane conjugated RTK derived pY peptide (~6000 /μm²), Rac1(GTP) (~4000 /μm²), and GβGγ (~4800 /μm²) by TIRF-M. Representative TIRF-M images showing the membrane localization of 20 nM nSH2-Cy3 in the absence and presence of membranes conjugated with a solution concentration of

*Figure 1 continued on next page*

*Figure 1 continued*

10 µM pY peptide. Representative images showing the membrane localization of 20 nM Cy3-p67/phox Rac1(GTP) sensor before (GDP) and after (GTP) the addition of the guanine nucleotide exchange factor, P-Rex1. Equilibrium localization of 50 nM (low) or 200 nM (high) farnesyl GβGγ-SNAP-AF488. (**E**) Representative TIRF-M images showing the equilibrium membrane localization of 10 nM Dy647-PI3Kγ measured in the absence and presence of membranes equilibrated with 200 nM farnesyl GβGγ. (**F**) Representative TIRF-M images showing the equilibrium membrane localization of 5 pM and 10 nM Dy647-PI3Kβ measured in the presence of membranes containing either pY, Rac1(GTP), or GβGγ. The inset image (+GβGγ) shows low-frequency single molecule binding events detected in the presence of 10 nM Dy647-PI3Kβ. Note that the contrast of the inset image was scaled differently to show the rare Dy647-PI3Kβ membrane binding events. (**G**) Bulk membrane absorption kinetics for 10 nM Dy647-PI3Kβ measured on membranes containing eitherpY, Rac1(GTP), or GβGγ. Membrane composition: 96% DOPC, 2% PI(4,5)$P_2$, 2% MCC-PE.

The online version of this article includes the following source data and figure supplement(s) for figure 1:

**Source data 1.** Related to *Figure 1C*.

**Source data 2.** Related to .*Figure 1G*.

**Figure supplement 1.** Characterization of Alexa488-SNAP-GβGγ localization on supported lipid bilayers.

**Figure supplement 1—source data 1.** Related to *Figure 1—figure supplement 1B*.

**Figure supplement 1—source data 2.** Related to *Figure 1—figure supplement 1C*.

**Figure supplement 2.** Characterization of Dy647-PI3Kβ membrane association with individual signaling inputs.

**Figure supplement 2—source data 1.** Related to *Figure 1—figure supplement 2A*.

**Figure supplement 2—source data 2.** Related to .*Figure 1—figure supplement 2B*.

**Figure supplement 2—source data 3.** Related to *Figure 1—figure supplement 2C*.

**Figure supplement 2—source data 4.** Related to .*Figure 1—figure supplement 2D*.

**Figure supplement 2—source data 5.** Related to .*Figure 1—figure supplement 2E*.

**Figure supplement 2—source data 6.** Related to .*Figure 1—figure supplement 2F*.

the low-frequency binding events we observed are mostly mediated by lipid interactions rather than direct binding to GβGγ (*Figure 1—figure supplement 2*). The addition of 20% PS lipids did, however, cause a subtle increase in the number of Dy647-PI3Kβ membrane binding observed when 20 µM pY peptide was added in solution (*Figure 1—figure supplement 2*). Our observations are consistent with previous reports involving the membrane binding dynamics of AF555-PI3Kα measured in the presence of solution pY peptide (*Ziemba et al., 2016*; *Buckles et al., 2017*). Given the large parameter space, we did not perform an exhaustive characterization of Dy647-PI3Kβ membrane binding across many membrane compositions. In this study, a simplified membrane composition was used to minimize non-specific membrane localization of fluorescently labeled PI3Kβ. This allowed us to more clearly define the strength of individual and combinations of protein-protein interactions that regulate PI3Kβ localization and kinase activity.

## PI3Kβ cooperatively engages a single membrane-tethered pY peptide

Previous biochemical analysis of PI3Kβ utilized pY peptides in solution to study the regulation of lipid kinase activity (*Zhang et al., 2011*; *Dbouk et al., 2012*). Using membrane-tethered pY peptide, we quantitatively mapped the relationship between the pY membrane surface density and the membrane binding behavior of Dy647-PI3Kβ (*Figure 2A*). To calculate the membrane surface density of conjugated pY, we incorporated a defined concentration of Alexa488-pY (*Figure 2A–B*). We measured the relationship between the total solution concentration of pY peptide used for the membrane conjugation step and the corresponding final membrane surface density (pY per µm²). Over a range of pY peptide solution concentrations (0–10 µM) used for maleimide coupling, we observed a linear increase in the membrane conjugation efficiency based on the incorporation of fluorescent Alexa488-pY (*Figure 2C*, **left hand y-axis**). Bulk membrane localization of a nSH2-Cy3 sensor showed a corresponding linear increase in fluorescence as a function of pY peptide membrane density (*Figure 2D*). By quantifying the average number of Alexa488-pY molecules per unit area on supported membranes, we calculated the absolute density of pY per µm² (*Figure 2C*, **right hand y-axis**).

To determine how the membrane binding behavior of PI3Kβ is modulated by the membrane surface density of pY, we measured the bulk membrane absorption kinetics of Dy647-PI3Kβ. When Dy647-PI3Kβ was flowed over membranes containing a surface density between 250 and 3000 pY/µm², we observed rapid equilibration kinetics consistent with a simple biomolecular interaction (*Figure 2E*).

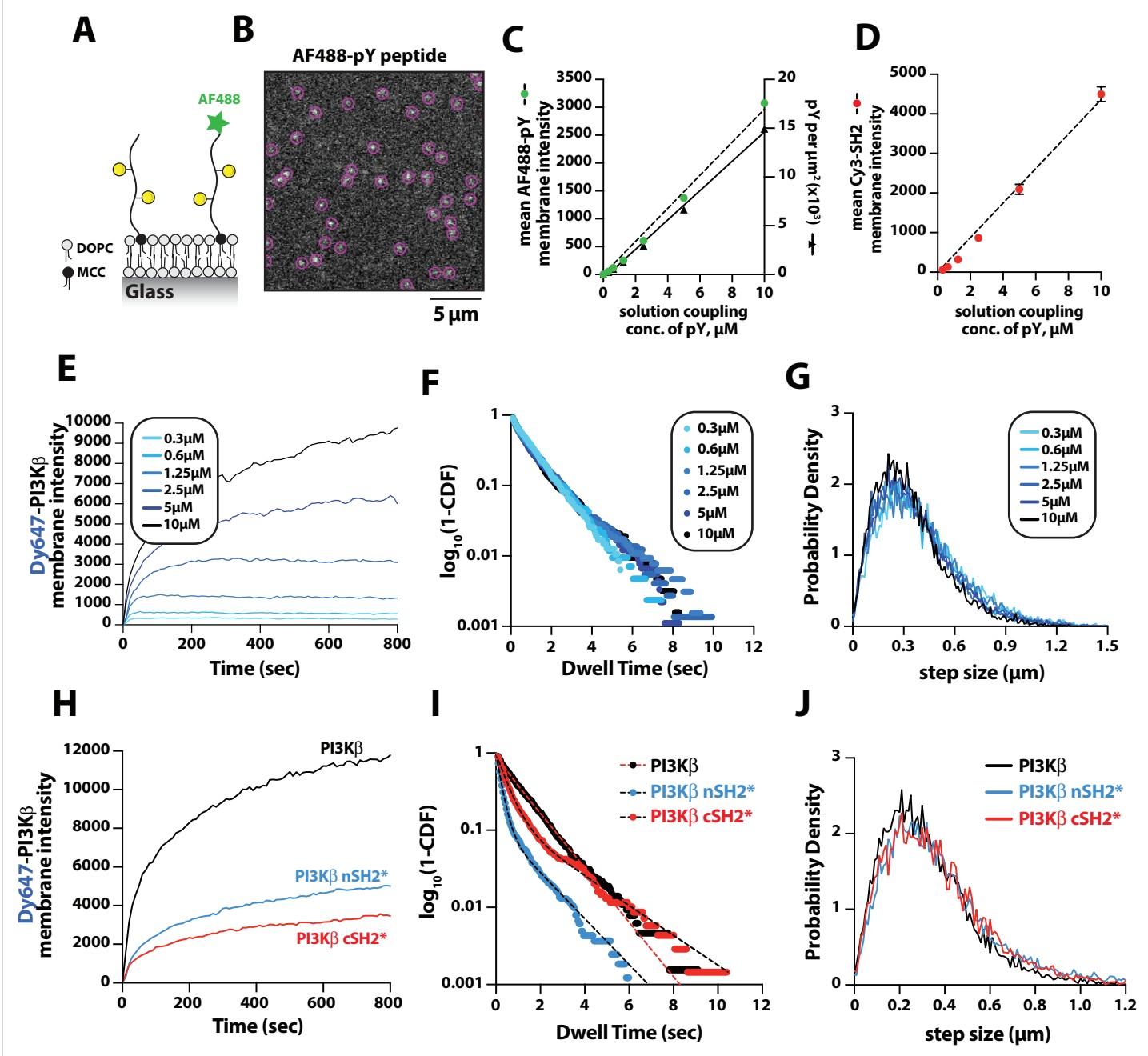

**Figure 2.** Density-dependent membrane binding behavior of Dy647-PI3Kβ measured in the presence of receptor tyrosine kinase (RTK)-derived phosphorylated (pY) peptides. (**A**) Cartoon schematic showing conjugation of pY peptides (+/- Alexa488 label) using thiol-reactive maleimide lipids (MCC-PE). (**B**) Representative image showing the single molecule localization of Alexa488-pY. Particle detection (purple circles) was used to quantify the number of pY peptides per μm². (**C**) Relationship between the total pY solution concentration (x-axis) used for covalent conjugation, the bulk membrane intensity of covalently attached Alexa488-pY (left y-axis), and the final surface density of pY peptides per μm² (right y-axis). (**D**) Relationship between the total pY solution conjugation concentration and bulk membrane intensity of measured in the presence of 50 nM nSH2-Cy3. (**E–G**) Membrane localization dynamics of Dy647-PI3Kβ measured on supported lipid bilayers (SLBs) containing a range of pY surface densities (250–15,000 pY/μm², based on *Figure 1C*). (**E**) Bulk membrane localization of 10 nM Dy647-PI3Kβ as a function of pY density. (**F**) Single molecule dwell time distributions measured in the presence of 5 pM Dy647-PI3Kβ. Data plotted as $\log_{10}(1-CDF)$ (cumulative distribution frequency). (**G**) Step size distributions showing Dy647-PI3Kβ single molecule displacements from >500 particles (>10,000 steps) per pY surface density. (**H–J**) Membrane localization dynamics of Dy647-PI3Kβ nSH2* (R358A) and cSH2* (R649A) mutants measured on SLBs containing ~15,000 pY/μm² (10 μM conjugation concentration). (**H**) Bulk membrane absorption kinetics of 10 nM Dy647-PI3Kβ (WT, nSH2*, and cSH2*). (**I**) Single molecule dwell time distributions measured in the presence of 5 pM Dy647-PI3Kβ (WT, nSH2*, and cSH2*). Data plotted as $\log_{10}(1-CDF)$ (cumulative distribution frequency). (**J**) Step size distributions showing single molecule displacements

*Figure 2 continued on next page*

*Figure 2 continued*

of >500 particles (>10,000 steps) in the presence of 5 pM Dy647-PI3Kβ (WT, nSH2*, and cSH2*). Membrane composition: 96% DOPC, 2% PI(4,5)P$_2$, 2% MCC-PE.

The online version of this article includes the following source data for figure 2:

**Source data 1.** Related to .*Figure 2C*.

**Source data 2.** Related to .*Figure 2D*.

**Source data 3.** Related to .*Figure 2E*.

**Source data 4.** Related to .*Figure 2F*.

**Source data 5.** Related to .*Figure 2G*.

**Source data 6.** Related to .*Figure 2H*.

**Source data 7.** Related to .*Figure 2I*.

**Source data 8.** Related to .*Figure 2J*.

Similar membrane binding kinetics have been reported for the Btk-PI(3,4,5)P$_3$ (*Chung et al., 2019*) and PI3Kγ-GβGγ (*Rathinaswamy et al., 2021*) complexes. When the membrane surface density was greater than 6500 pY/μm$^2$, we observed slower equilibration kinetics consistent with a fraction of Dy647-PI3Kβ complexes exhibiting longer dwell times. Single particle tracking of Dy647-PI3Kβ on membranes containing varying densities of pY peptide revealed that the dwell time was relatively insensitive to the pY peptide density (*Figure 2F* and *Table 1*). However, we could be underestimating the actual dwell time of Dy647-PI3Kβ due to membrane hopping (*Yasui et al., 2014*). In the presence of a high pY surface density, Dy647-PI3Kβ could dissociate from pY and then immediately rebind to another pY instead of diffusing away into the solution phase. Quantification of single particle displacement (or step size) of pY-tethered Dy647-PI3Kβ complexes revealed nearly identical diffusivity across a range of pY membrane densities (*Figure 2G* and *Table 1*). Together, these results suggest that Dy647-PI3Kβ most frequently engages a single dually phosphorylated peptide over a broad range of pY densities in our bilayer assay.

The regulatory subunit of PI3Kβ (p85α) contains two SH2 domains that form inhibitory contacts with the catalytic domain (p110β) (*Zhang et al., 2011*). The SH2 domains of class 1A PI3Ks have a conserved peptide motif, FLVR, that mediates the interaction with tyrosine phosphorylated peptides (*Bradshaw et al., 1999*; *Waksman et al., 1992*; *Rameh et al., 1995*). Mutating the arginine to alanine (FLVA mutant) prevents the interaction with pY peptides for both PI3Kα and PI3Kβ (*Yu et al., 1998*; *Dornan et al., 2020*; *Nolte et al., 1996*; *Zhang et al., 2011*; *Breeze et al., 1996*). To determine how

**Table 1.** Diffusion coefficients of Dy647-PI3Kβ (WT and SH2 domain mutants) bound to membrane-tethered tyrosine phosphorylated peptides.

| protein visualized | pY/μm$^2$ | $\tau_1 \pm SD$ (s) | $\tau_2 \pm SD$ (s) | $\alpha(\tau) \pm SD$ | AVE DT (s) | $D_1 \pm SD$ (μm$^2$/sec) | $D_2 \pm SD$ (μm$^2$/s) | $\alpha(_D) \pm SD$ | MEDIAN step (μm) |
|---|---|---|---|---|---|---|---|---|---|
| PI3Kβ (WT) | 250 | 0.58±0.28 | 1.78±0.58 | 0.60±0.37 | 1.00±0.09 | 0.39±0.07 | 1.45±0.09 | 0.29±0.08 | 0.37±0.02 |
| PI3Kβ (WT) | 573 | 0.39±0.06 | 1.37±0.14 | 0.27±0.02 | 1.12±0.09 | 0.28±0.06 | 1.15±0.14 | 0.22±0.04 | 0.35±0.01 |
| PI3Kβ (WT) | 1226 | 0.36±0.13 | 1.29±0.06 | 0.30±0.09 | 1.05±0.11 | 0.20±0.02 | 1.18±0.09 | 0.16±0.02 | 0.36±0.02 |
| PI3Kβ (WT) | 2935 | 0.44±0.11 | 1.53±0.38 | 0.47±0.25 | 1.00±0.07 | 0.28±0.09 | 1.09±0.12 | 0.26±0.09 | 0.33±0.01 |
| PI3Kβ (WT) | 6661 | 0.46±0.08 | 1.28±0.16 | 0.61±0.13 | 0.82±0.11 | 0.35±0.17 | 1.28±0.34 | 0.35±0.18 | 0.34±0.01 |
| PI3Kβ (WT) | 14944 | 0.55±0.11 | 1.44±0.56 | 0.54±0.22 | 0.91±0.09 | 0.45±0.15 | 1.40±0.54 | 0.48±0.06 | 0.33±0.05 |
| PI3Kβ (WT) | 14944 | 0.49±0.17 | 1.38±0.19 | 0.35±0.17 | 1.10±0.04 | 0.32±0.04 | 0.99±0.12 | 0.40±0.11 | 0.30±0.02 |
| PI3Kβ (nSH2*) | 14944 | 0.23±0.02 | 1.48±0.23 | 0.86±0.03 | 0.45±0.06 | 0.38±0.09 | 1.45±0.25 | 0.41±0.18 | 0.34±0.01 |
| PI3Kβ (cSH2*) | 14944 | 0.38±0.08 | 1.54±0.55 | 0.76±0.1 | 0.65±0.08 | 0.34±0.13 | 1.12±0.29 | 0.43±0.15 | 0.30±0.04 |

SD = standard deviation from 3-5 technical replicates. n = 331 – 1909 total particles for each technical replicate. steps = 4277 – 39378 total particle displacements measured for each technical replicate. Alpha (α$_t$) equals the fraction of molecules with the characteristic dwell time, $\tau_1$ (DT = dwell time). The fraction of molecules with the characteristic dwell time, $\tau_2$, equals 1-α$_t$. Alpha (α$_D$) equals the fraction of molecules with the characteristic diffusion coefficient, $D_1$. The fraction of molecules with diffusion coefficient, $D_2$, equals 1-α$_D$. Membrane composition: 96% DOPC, 2% PI(4,5)P$_2$, 2% MCC-PE.

the membrane binding behavior of PI3Kβ is modulated by each SH2 domain, we individually mutated the FLVR amino acid sequence to FLVA. Compared to wild-type Dy647-PI3Kβ, the nSH2(R358A) and cSH2(R649A) mutants showed a 60% and 75% reduction in membrane localization at equilibrium, respectively (*Figure 2H*). Single molecule dwell time analysis also showed a significant reduction in membrane affinity for Dy647-PI3Kβ nSH2(R358A) and cSH2(R649A) compared to wild-type PI3Kβ (*Figure 2I* and *Table 1*). Single molecule diffusion (or mobility) of membrane-bound nSH2(R358A) and cSH2(R649A) mutants, however, were nearly identical to wild-type Dy647-PI3Kβ (*Figure 2J* and *Table 1*). Because the nSH2 and cSH2 mutants can only interact with a single phosphorylated tyrosine residue on the doubly phosphorylated pY peptide, this further supports a model in which the p85α regulatory subunit of PI3Kβ cooperatively engages one doubly phosphorylated pY peptide under our experimental conditions.

## GβGγ-dependent enhancement in PI3Kβ localization requires the release of the nSH2

Having established that PI3Kβ engagement with a membrane-tethered pY peptide is the critical first step for robust membrane localization, we examined the secondary role that GβGγ serves in controlling membrane localization of PI3Kβ bound to pY. To measure synergistic membrane localization mediated by the combination of pY and GβGγ, we covalently linked pY peptides to the supported membrane at a surface density of ~5000 pY/μm$^2$ and then allowed farnesyl GβGγ to equilibrate into the membrane to density of ~4700 GβGγ/μm$^2$. We quantified the membrane surface density of GβGγ at equilibrium using a combination of AF488-SNAP-GβGγ and dilute AF555-SNAP-GβGγ (0.0025%), to visualize the bulk and single molecule densities, respectively (*Figure 3A*). Comparing the bulk membrane absorption of Dy647-PI3Kβ in the presence of pY alone, we observed a two-fold increase in membrane localization due to synergistic association with pY and GβGγ (*Figure 3B–C*). Single molecule imaging experiments also showed a 1.9-fold increase in the membrane dwell time of Dy647-PI3Kβ in the presence of both pY and GβGγ (*Figure 3D*). Consistent with Dy647-PI3Kβ forming a complex with pY and GβGγ, we observed a 22% reduction in the average single-particle displacement and a decrease in the diffusion coefficient due to ternary complex formation (*Figure 3E*).

Parallel to our experiments using membrane-conjugated pY, we examined whether solution pY could promote Dy647-PI3Kβ localization to GβGγ-containing membranes. Based on the bulk membrane recruitment, solution pY did not strongly enhance membrane binding of Dy647-PI3Kβ on GβGγ-containing membranes in the absence of phosphatidylserine (PS) lipids (*Figure 3B*). Single molecule dwell time analysis revealed few transient Dy647-PI3Kβ membrane interactions (inset *Figure 3A*) with a mean dwell time of 116 ms in the presence of GβGγ alone (*Figure 3—figure supplement 1A*). The presence of 10 μM solution pY modestly increased the mean dwell time of Dy647-PI3Kβ to 136 ms on GβGγ containing membranes (*Figure 3—figure supplement 1B*). When we measured the interaction between Dy647-PI3Kβ and GβGγ on a membrane containing 20% PS, however, we observed a significant enhancement in Dy647-PI3Kβ localization when solution pY was added (*Figure 3—figure supplement 1C–1D*). This confirms that SH2 domain-dependent inhibition of GβGγ binding can be relieved by solution pY, as previously reported (*Dbouk et al., 2012*). However, a high density of anionic lipids (i.e. PS) was required to facilitate the interaction between pY and GβGγ. This suggests that the affinity between PI3Kβ and GβGγ is relatively weak, which is consistent with previous structural biochemistry studies (*Dbouk et al., 2012*).

For PI3Kβ to engage GβGγ, it is hypothesized that the nSH2 domain must move out of the way from sterically occluding the GβGγ binding site. This model is supported by previous hydrogen-deuterium exchange mass spectrometry (HDX-MS) experiments that could only detect interactions between GβGγ and PI3Kβ (p110β) when the nSH2 domain was either deleted or disengaged from the catalytic domain by a soluble RTK-derived pY peptide (*Dbouk et al., 2012*). We examined the putative interface of GβGγ bound to the p110β catalytic domain using AlphaFold multimer (*Jumper et al., 2021*; *Evans et al., 2022*; *Varadi et al., 2022*) which defined hα1 in the helical domain as the binding site. This result was consistent with previous mutagenesis and HDX-MS analysis of GβGγ binding to p110β (*Dbouk et al., 2012*). Comparing our model to previous X-ray crystallographic data of SH2 binding to either p110α and p110β (*Zhang et al., 2011*; *Mandelker et al., 2009*) suggested that the nSH2 domain sterically obstructs the GβGγ binding interface (*Figure 3F* and *Figure 3—figure supplement 2*), with GβGγ activation only possible when the nSH2 dissociates from p110β interface. To

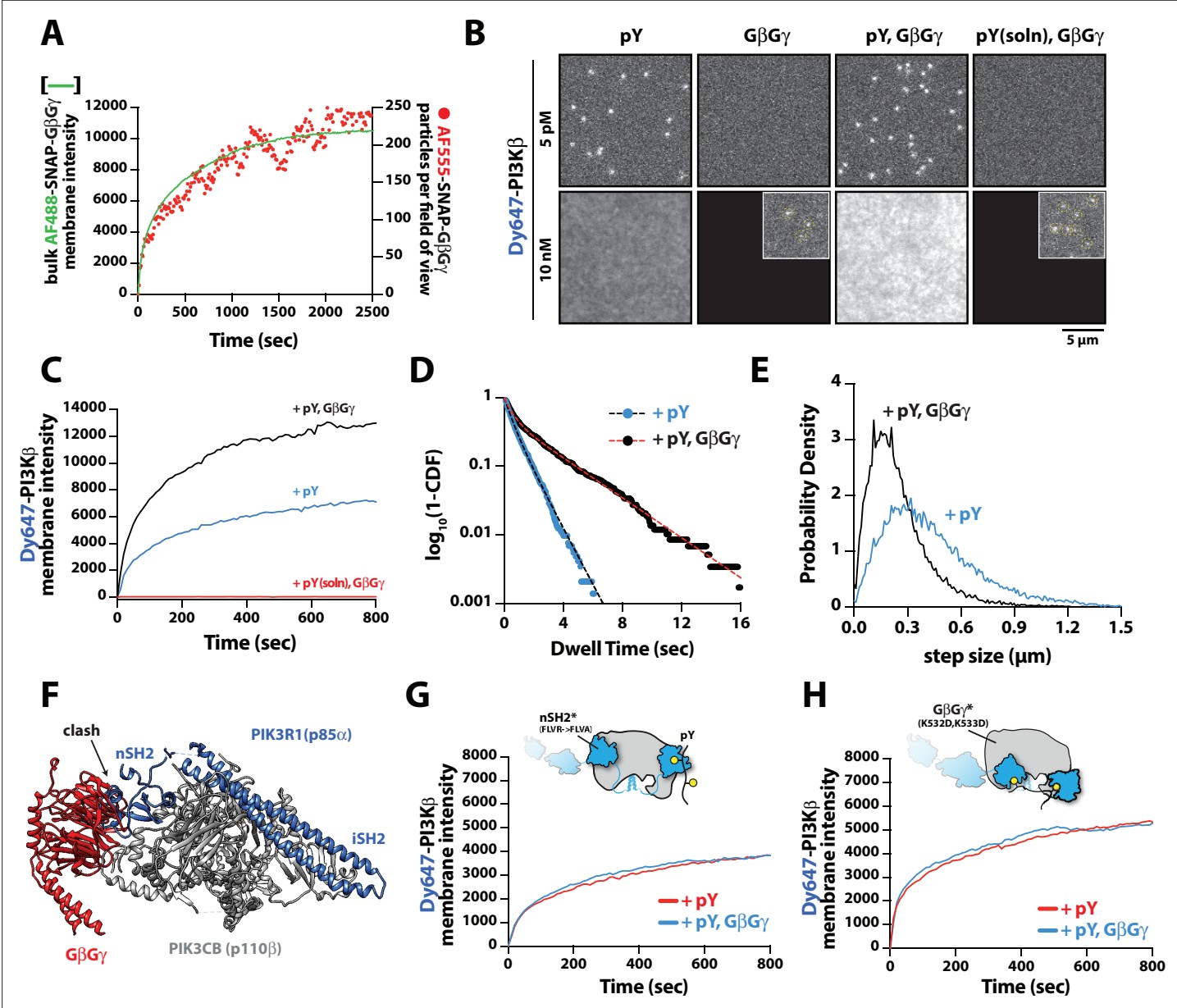

**Figure 3.** Mechanism controlling synergistic Dy647-PI3Kβ membrane binding by phosphorylated (pY) and G-protein complexes (GβGγ). (**A**) Kinetic trace showing the membrane absorption of 200 nM AF488-SNAP-GβGγ containing 0.0025% AF555-SNAP-GβGγ measured by TIRF-M. Single molecule densities of AF555-SNAP-GβGγ were calculated for each frame in a field of view of 3000 µm². (**B**) Representative TIRF-M images showing the equilibrium membrane localization of 5 pM and 10 nM Dy647-PI3Kβ on membranes containing either pY, GβGγ, pY/GβGγ, or pY(solution)/GβGγ. The inset image (+GβGγ and +pY/GβGγ) shows low-frequency single molecule binding events detected in the presence of 10 nM Dy647-PI3Kβ. Supported membranes were conjugated with 10 µM pY peptide (final surface density of ~15,000 pY/µm²) and equilibrated with 200 nM farnesyl-GβGγ before adding Dy647-PI3Kβ. pY (solution)=10 µM. (**C**) Bulk membrane recruitment dynamics of 10 nM Dy647-PI3Kβ measured in the presence of either pY alone, pY/GβGγ, or pY (solution)/GβGγ. pY(solution)=10 µM. (**D**) Single molecule dwell time distributions measured in the presence of 5 pM Dy647-PI3Kβ on supported membranes containing pY alone ($\tau_1$=0.55 ± 0.11 s, $\tau_2$=1.44 ± 0.56 s, α=0.54, n=4698 particles, n=5 technical replicates) or pY/GβGγ ($\tau_1$=0.61 ± 0.13 s, $\tau_2$=3.09 ± 0.27 s, α=0.58, n=3421 particles, n=4 technical replicates). Alpha(α) represents the fraction of particles characterized by the time constant ($\tau_1$). (**E**) Step size distributions showing single molecule displacements measured in the presence of either pY alone (D1=0.34 ± 0.04 µm²/s, D2=1.02 ± 0.07 µm²/s, α=0.45) or pY/GβGγ (D1=0.23 ± 0.03 µm²/s, D2=0.88 ± 0.08 µm²/s,α=0.6); n=3-4 technical replicates from >3000 tracked particles with 10,000-30,000 total displacements measured. Alpha(α) represents the fraction of particles characterized by the diffusion coefficient (D1). (**F**) Combined model of the putative nSH2 and GβGγ binding sites on p110β. The p110β-GβGγ binding site is based on an Alphafold multimer model supported by previous HDX-MS and mutagenesis experiments. The orientation of the nSH2 is based on previous X-ray crystallographic data on PI3Kα (p110α-p85α, niSH2, PDB:3HHM) aligned to the structure of PI3Kβ (p110β-p85α, icSH2, PDB:2Y3A). (**G**) Bulk membrane recruitment dynamics of 10 nM Dy647-PI3Kβ, WT and nSH2(R358A), measured on membranes containing either pY or pY/GβGγ. (**H**) Bulk membrane recruitment dynamics of 10 nM Dy647-PI3Kβ, WT

*Figure 3 continued on next page*

*Figure 3 continued*

and GβGγ binding mutant, measured on membranes containing either pY or pY/GβGγ. (**A–H**) Membrane composition: 96% DOPC, 2% PI(4,5)P$_2$, 2% MCC-PE.

The online version of this article includes the following source data and figure supplement(s) for figure 3:

**Source data 1.** Related to *Figure 3A*.

**Source data 2.** Related to *Figure 3C*.

**Source data 3.** Related to *Figure 3D*.

**Source data 4.** Related to *Figure 3E*.

**Source data 5.** Related to *Figure 3G*.

**Source data 6.** Related to *Figure 3H*.

**Figure supplement 1.** Solution phosphorylated (pY) peptide slightly enhances Dy647-PI3Kβ binding to G-protein complexes (GβGγ) membranes.

**Figure supplement 1—source data 1.** Related to Fig.*Figure 3—figure supplement 1A*.

**Figure supplement 1—source data 2.** Related to *Figure 3—figure supplement 1B*.

**Figure supplement 1—source data 3.** Related to *Figure 3—figure supplement 1D*.

**Figure supplement 2.** Structural model for G-protein complexes (GβGγ) binding to the Phosphoinositide 3-kinase beta (PI3Kβ) catalytic subunit (p110β).

test this hypothesis, we measured the membrane binding dynamics of Dy647-PI3Kβ nSH2(R358A) on membranes containing pY and GβGγ. Comparing the bulk membrane recruitment of these constructs revealed that the inability of the Dy647-PI3Kβ nSH2 domain to bind to pY peptides made the kinase insensitive to synergistic membrane recruitment mediated by pY and GβGγ (*Figure 3G*). Similarly, the membrane association dynamics of Dy647-PI3Kβ nSH2(R358A), phenocopied a PI3Kβ (K532D, K533D) mutant that lacks the ability to engage GβGγ (*Figure 3H*).

## Rac1(GTP) and pY synergistically enhance PI3Kβ membrane localization

PI3Kβ is the only class IA PI3K shown to interact with Rho-family GTPases, Rac1, and Cdc42 (*Fritsch et al., 2013*). Our membrane localization studies indicate, however, that Dy647-PI3Kβ does not strongly localize to membranes containing Rac1(GTP) alone (*Figure 1C–D* and *Figure 1—figure supplement 2B*). To determine whether membrane-anchored pY peptides can facilitate interactions with Rac1(GTP), we visualized the localization of Dy647-PI3Kβ on membranes containing pY-Rac1(GDP) or pY-Rac1(GTP). Our experiments were designed to have the same pY surface density across conditions. By incorporating a small fraction of Cy3-Rac1 and Alexa488-pY into our Rac1-pY membrane coupling reaction we were able to visualize single membrane-anchored proteins and calculate the membrane surface density of ~4000 Rac1/μm$^2$ and ~5000 pY/μm$^2$ (*Figure 4A–B*). Bulk localization to membranes containing either pY-Rac1(GDP) or pY-Rac1(GTP), revealed that active Rac1 could enhance Dy647-PI3Kβ localization 1.4-fold (*Figure 4C–D*). Similarly, single molecule analysis revealed a 1.5-fold increase in the mean dwell time of Dy647-PI3Kβ in the presence of pY-Rac1(GTP) (*Figure 4E*). The average displacement of Dy647-PI3Kβ per frame (i.e. 52 ms) also decreased by 28% in the presence of pY and Rac1(GTP) (*Figure 4F*), consistent with the formation of a membrane-bound PI3Kβ-pY-Rac1(GTP) ternary complex.

## Rac1(GTP) and GβGγ stimulate PI3Kβ activity beyond enhancing membrane localization

Previous in vitro measurements of PI3Kβ activity have shown that solution pY stimulates lipid kinase activity (*Zhang et al., 2011*; *Dbouk et al., 2012*). Similar mechanisms of activation have been reported for other class IA kinases, including PI3Kα and PI3Kδ (*Buckles et al., 2017*; *Burke et al., 2011*; *Dornan et al., 2017*). Functioning in concert with pY peptides, GβGγ, or Rho-family GTPase synergistically enhance PI3Kβ activity by a mechanism that remains unclear (*Fritsch et al., 2013*; *Dbouk et al., 2012*). Similarly, RTK-derived peptides and H-Ras (GTP) have been shown to synergistically activate PI3Kα (*Buckles et al., 2017*; *Siempelkamp et al., 2017*; *Yang et al., 2012*). In the case of PI3Kβ, previous experiments have not determined whether synergistic activation by multiple signaling inputs results from an increase in membrane affinity ($K_D$) or direct modulation of lipid kinase activity ($k_{cat}$) through an allosteric mechanism. To determine how the lipid kinase activity of the PI3Kβ-pY complex is synergistically modulated by either GβGγ or Rho-family GTPases, we used TIRF-M to

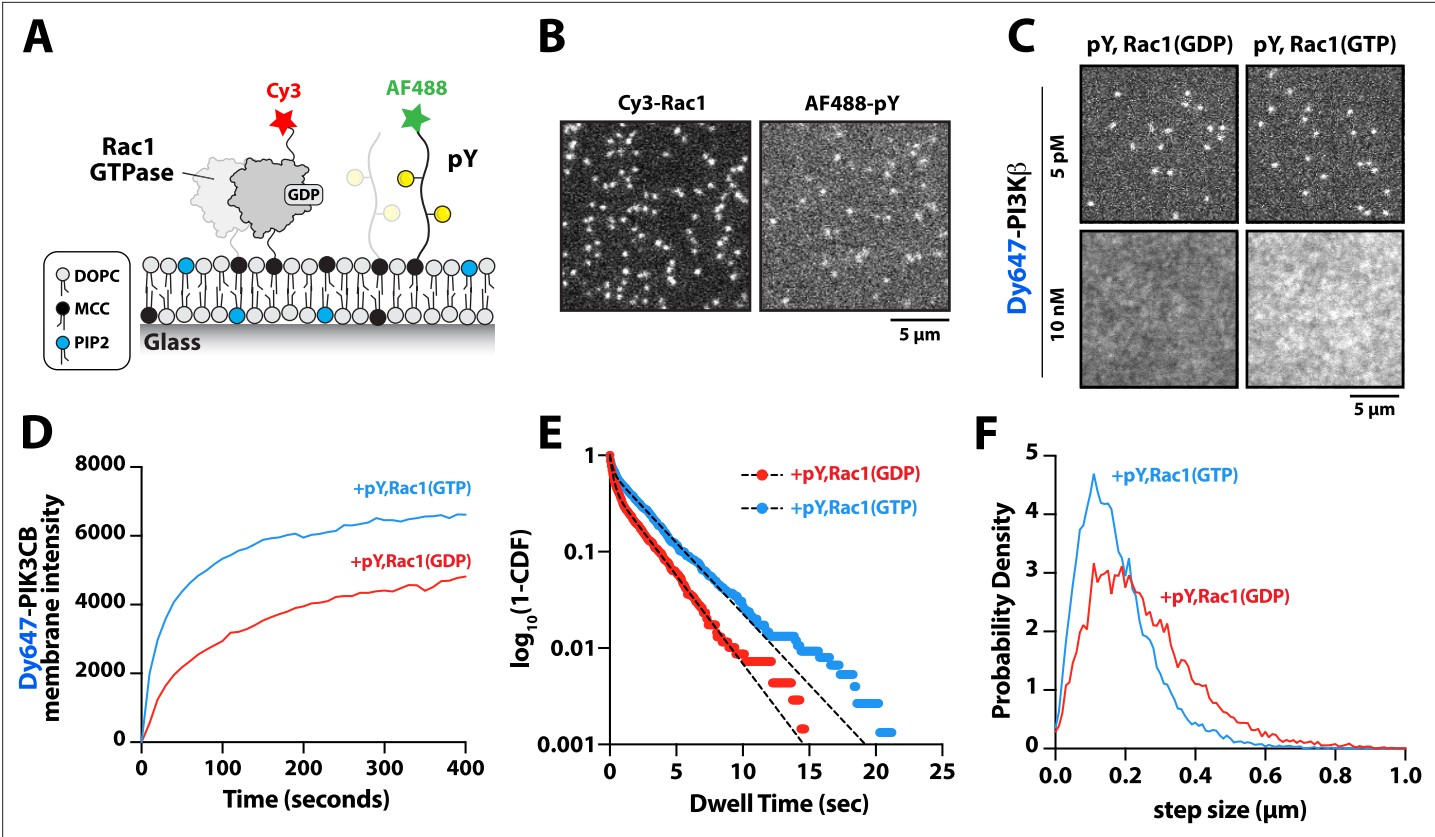

**Figure 4.** Membrane-anchored phosphorylated (pY) peptides synergistically enhance Dy647-PI3Kβ membrane binding in the presence of Rac1(GTP). (**A**) Cartoon schematic showing membrane conjugation of Cy3-Rac1 and AF488-pY on membranes containing unlabeled Rac1 and pY. (**B**) Representative TIRF-M images showing localization of Cy3-Rac1 (1:10,000 dilution) and AF488-pY (1:30,000 dilution) after membrane conjugation in the presence of 30 μM Rac1 and 10 μM pY. Membrane surface density equals ~4000 Rac1/μm² and ~5000 pY/μm².(**C**) Representative TIRF-M images showing the equilibrium membrane localization of 5 pM and 10 nM Dy647-PI3Kβ measured in the presence of membranes containing either pY/Rac1(GDP) or pY/Rac1(GTP). (**D** ) Bulk membrane recruitment dynamics of 10 nM Dy647-PI3Kβ measured in the presence of pY/Rac1(GDP) or pY/Rac1(GTP). (**E**) Single molecule dwell time distributions measured in the presence of 5 pM Dy647-PI3Kβ on supported membranes containing pY/Rac1(GDP) or pY/Rac1(GTP). (**F**) Step size distributions showing single molecule displacements from >500 Dy647-PI3Kβ particles (>10,000 steps) in the presence of either pY/Rac1(GDP) or pY/Rac1(GTP). Membrane composition: 96% DOPC, 2% PI(4,5)P₂, 2% MCC-PE.

The online version of this article includes the following source data for figure 4:

**Source data 1.** Related to *Figure 4D*.

**Source data 2.** Related to *Figure 4E*.

**Source data 3.** Related to *Figure 4F*.

simultaneously visualize Dy647-PI3Kβ membrane localization and monitor PI(3,4,5)P₃ production. To measure the kinetics of PI(3,4,5)P₃ formation, we purified and fluorescently labeled the pleckstrin homology and Tec homology (PH-TH) domain derived from Bruton's tyrosine kinase (Btk). We used a form of Btk containing a mutation that disrupts the peripheral PI(3,4,5)P₃ lipid binding domain (*Wang et al., 2015*). This Btk mutant was previously shown to associate with a single PI(3,4,5)P₃ head group and exhibits rapid membrane equilibration kinetics in vitro (*Chung et al., 2019*). Consistent with previous observations, Btk fused to SNAP-AF488 displayed high specificity and rapid membrane equilibration kinetics on SLBs containing PI(3,4,5)P₃ (*Figure 5A, B*). Compared to the PI(3,4,5)P₃ lipid sensor derived from the Cytohesin/Grp1 PH domain (*He et al., 2008*; *Knight et al., 2010*), the Btk lipid sensor exhibited a faster association rate constant ($k_{ON}$) and a more transient dwell time ($1/k_{OFF}$) making it ideal for kinetic analysis of PI3Kβ lipid kinase activity (*Figure 5—figure supplement 1*). Using Btk-SNAP-AF488, we measured the production of PI(3,4,5)P₃ lipids on SLBs by quantifying the time-dependent recruitment in the presence of PI3Kβ. The change in Btk-SNAP-AF488 membrane

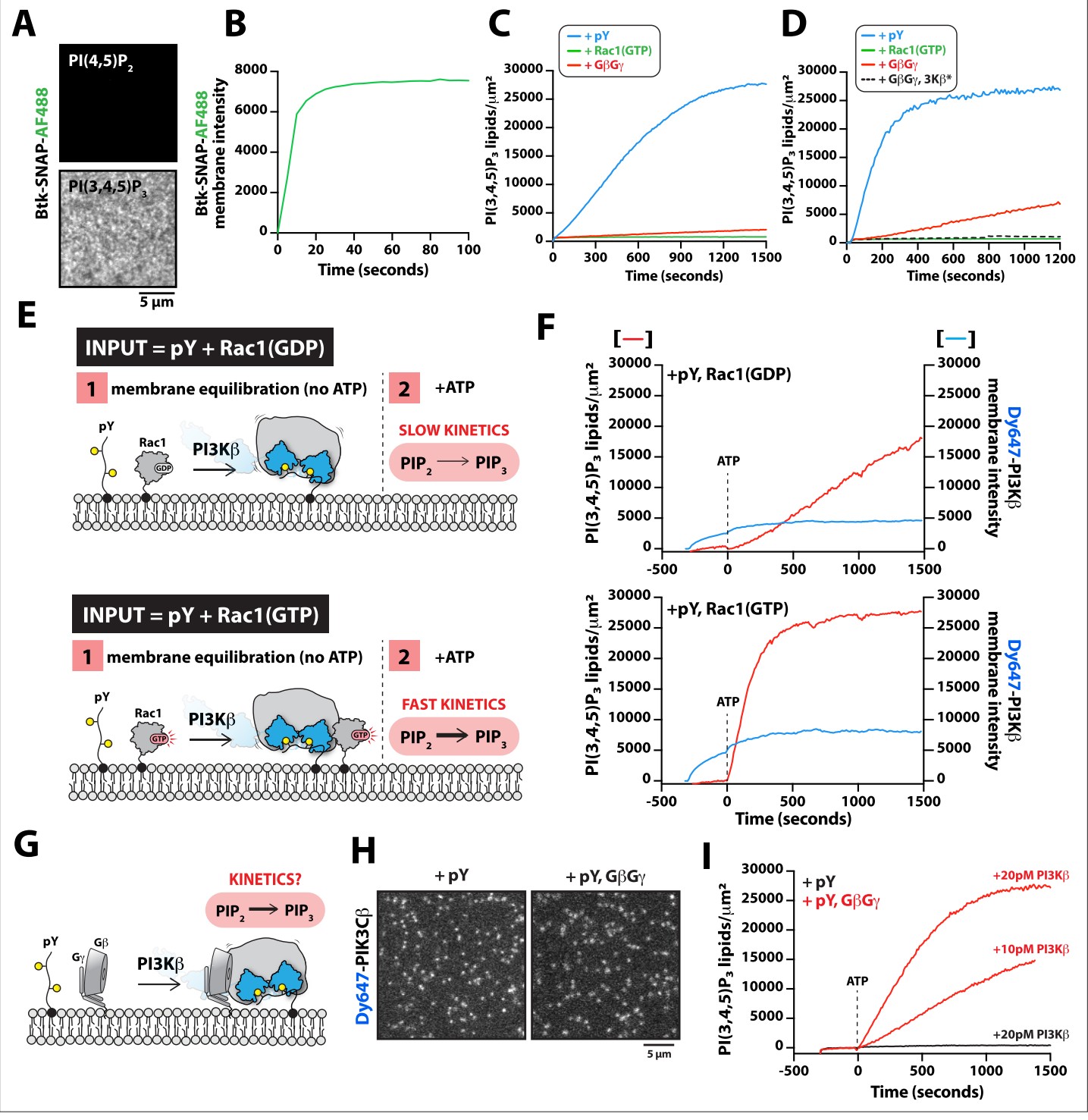

**Figure 5.** G-protein complexes (GβGγ) and Rac1(GTP) stimulate phosphoinositide 3-kinase beta (PI3Kβ) activity beyond enhancing localization on phosphorylated (pY) membranes. (**A**) Representative TIRF-M images showing localization of 20 nM Btk-SNAP-AF488 on supported lipid bilayers (SLBs) containing either 2% PI(4,5)P$_2$ or 2% PI(3,4,5)P$_3$, plus 98% DOPC. (**B**) Bulk membrane recruitment kinetics of 20 nM Btk-SNAP-AF488 on an SLB measured by TIRF-M. (**C–D**) Kinetics of PI(3,4,5)P$_3$ production measured in the presence of 10 nM Dy647-PI3Kβ and 1 mM ATP on SLBs with membrane anchored pY, Rac1(GTP), or GβGγ alone. Reactions in (**C**) were performed in the absence of PS lipids, while membranes in (**D**) contained 20% DOPS. (**E**) Cartoon schematic illustrating method for measuring Dy647-PI3Kβ activity in the presence of either pY/Rac1(GDP) or pY/Rac1(GTP). Phase 1 of the reconstitution involves membrane equilibration of Dy647-PI3Kβ in the absence of ATP. During phase 2, 1 mM ATP was added to stimulate lipid kinase activity of Dy647-PI3Kβ. (**F**) Dual color TIRF-M imaging showing 2 nM Dy647-PI3Kβ localization and catalysis measured in the presence of 20 nM Btk-SNAP-AF488. Dashed line represents the addition of 1 mM ATP to the reaction chamber. (**G**) Cartoon schematic showing experimental design for measuring synergistic

*Figure 5 continued on next page*

*Figure 5 continued*

binding and activation of Dy647-PI3Kβ in the presence of pY and GβGγ. (**H**) Representative single molecule TIRF-M images showing the localization of 20 pM Dy647-PI3Kβ in (**G**). (**I**) Kinetics of PI(3,4,5)P₃ production monitored in the presence of 20 nM Btk-SNAP-AF488 and 10-20 pM Dy647-PI3Kβ. Membrane contained either pY or pY/GβGγ. (**B, C, F, H, I**) Membrane composition: 96% DOPC, 2% PI(4,5)P₂, 2% MCC-PE. (**D**) Membrane composition: 76% DOPC, 20% DOPS, 2% PI(4,5)P₂, 2% MCC-PE. All kinetic measurements of PI(3,4,5)P₃ production were performed in the presence of 20 nM Btk-SNAP-AF488.

The online version of this article includes the following source data and figure supplement(s) for figure 5:

**Source data 1.** Related to *Figure 5B*.

**Source data 2.** Related to *Figure 5C*.

**Source data 3.** Related to *Figure 5D*.

**Source data 4.** Related to *Figure 5F*.

**Source data 5.** Related to *Figure 5I*.

**Figure supplement 1.** Characterization of Bruton's tyrosine kinase (Btk) and Grp1 membrane binding.

**Figure supplement 1—source data 1.** Related to *Figure 5—figure supplement 1B*.

**Figure supplement 1—source data 2.** Related to *Figure 5—figure supplement 1C*.

**Figure supplement 1—source data 3.** Related to *Figure 5—figure supplement 1D*.

**Figure supplement 1—source data 4.** Related to *Figure 5—figure supplement 1E*.

**Figure supplement 1—source data 5.** Related to *Figure 5—figure supplement 1F*.

**Figure supplement 2.** Reconstitution of Phosphoinositide 3-kinase beta (PI3Kβ) lipid kinase activity on supported membranes.

**Figure supplement 2—source data 1.** Related to .*Figure 5—figure supplement 2B*.

**Figure supplement 2—source data 2.** Related to *Figure 5—figure supplement 2A*.

**Figure supplement 2—source data 3.** Related to *Figure 5—figure supplement 2B*.

fluorescence could be converted to the absolute number of PI(3,4,5)P₃ lipids produced per µm² to determine the catalytic efficiency per membrane-bound Dy647-PI3Kβ.

While our binding experiments provide a mechanism for enhanced PI3Kβ membrane localization in the presence of either pY-Rac1(GTP) or pY-GβGγ, these results did not reveal the mechanism controlling synergistic activation of lipid kinase activity. To probe if synergistic activation results from enhanced membrane localization or allosteric modulation of PI3Kβ, we first examined how well the pY peptide stimulates PI3Kβ lipid kinase activity on SLBs. In the absence of pY peptides, PI3Kβ did not catalyze the production of PI(3,4,5)P₃ lipids, while the addition of 10 µM pY in solution resulted in a subtle increase in PI3Kβ lipid kinase activity (*Figure 5—figure supplement 2A–2C*). By contrast, covalent conjugation of pY peptides to supported lipid bilayers increased the rate of PI(3,4,5)P₃ production by 207-fold (*Figure 5—figure supplement 2A–2C*). The observed difference in kinetics was consistent with robust membrane recruitment of Dy647-PI3Kβ requiring membrane-tethered pY peptides. Comparing the lipid kinase activity of PI3Kβ measured in the presence of individual signaling inputs – pY, Rac1(GTP), and GβGγ – revealed that pY alone provided the strongest stimulation (*Figure 5C*). This was largely due to the dramatic enhancement in membrane recruitment caused by membrane-tethered pY peptides. Incorporation of 20% DOPS lipids in the SLBs raised the overall activity of PI3Kβ across all conditions but the general trend for signaling input preference was preserved (*Figure 5D*). Similar to previous observations (*Dbouk et al., 2012*), we found that GβGγ alone could subtly stimulate PI3Kβ lipid kinase activity, without strongly enhancing membrane PI3Kβ localization (*Figure 1—figure supplement 2*). By contrast, the PI3Kβ (K532D, K533D) mutant that was unable to interact with GβGγ was insensitive to GβGγ-mediated activation (*Figure 5D*).

Next, we sought to test whether the combination of pY and Rac1(GTP) could synergistically stimulate PI3Kβ activity beyond the expected increase due to the enhanced membrane localization of the PI3Kβ-pY-Rac1(GTP) complex. To decipher the mechanism of synergistic activation, we performed two-phase experiments that accounted for both the total density of membrane bound Dy647-PI3Kβ and the corresponding kinetics of PI(3,4,5)P₃ generation. In phase 1 of our experiments, Dy647-PI3Kβ was flowed over SLBs and allowed to equilibrate with either pY-Rac1(GDP) or pY-Rac1(GTP) in the absence of ATP (*Figure 5E, F*). This resulted in a 1.8-fold increase in Dy647-PI3Kβ localization mediated by the combination of membrane-tethered Rac1(GTP) and pY, compared to pY membranes

alone. Following membrane equilibration of Dy647-PI3Kβ, phase 2 was initiated by adding a final concentration of 1 mM ATP to the reaction chamber to stimulate lipid kinase activity. We found that the addition of ATP did not alter the bulk localization of Dy647-PI3Kβ, though the kinase was in dynamic equilibrium between the solution and membrane. Conducting experiments in this manner allowed us to measure activation by inputs while removing uncertainty from differential Dy647-PI3Kβ association with various signaling inputs. After accounting for the 1.8-fold difference in Dy647-PI3Kβ membrane localization comparing pY-Rac1(GDP) and pY-Rac1(GTP) membranes, we calculated a 4.3-fold increase in PI3Kβ activity that was dependent on Rac1(GTP).

Using the two-phase kinase assay described above, we next examined how pY and GβGγ synergistically activate PI3Kβ (*Figure 5G*). In our pilot experiments, we immediately observed more robust activation of PI3Kβ in the presence of pY-GβGγ, compared to pY-Rac1(GTP). To accurately measure the rapid kinetics of PI(3,4,5)P$_3$ generation on SLBs we had to use a 100-fold lower concentration of Dy647-PI3Kβ. Under these conditions, single membrane-bound Dy647-PI3Kβ molecules could be spatially resolved, which allowed us to measure the catalytic efficiency per PI3Kβ (*Figure 5H*). Comparing the activity of Dy647-PI3Kβ on membranes with pY alone or pY-GβGγ, we observed a 22-fold increase in catalytic efficiency comparing the PI3Kβ-pY and PI3Kβ-pY-GβGγ complexes (*Figure 5I*). Synergistic activation was dependent on the direct interaction between PI3Kβ and GβGγ (*Figure 5—figure supplement 2D*). By varying the density of membrane-anchored GβGγ, we determined that maximum synergistic activation occurred when GβGγ was present at a density greater than 2400 molecules/μm$^2$ (*Figure 5—figure supplement 2D*). Similar levels of PI3Kβ activity were measured in the ATP spike experiments when 20% DOPS was incorporated in the supported membranes (*Figure 5—figure supplement 2E*). Based on the membrane-bound density of ~0.2 Dy647-PI3Kβ molecules per μm$^2$, we calculate a $k_{cat}$ of 57 PI(3,4,5)P$_3$ lipids/s•PI3Kβ on pY-GβGγ containing membranes. By contrast, the Dy647-PI3Kβ-pY complex had a $k_{cat}$ of 3 PI(3,4,5)P$_3$ lipids/s•PI3Kβ.

## Discussion

### Prioritization of signaling inputs

The exact mechanisms that regulate how PI3Kβ prioritizes interactions with signaling input, such as pY, Rac1(GTP), and GβGγ remains unclear. To fill this gap in knowledge, we directly visualized the membrane association and dissociation dynamics of fluorescently labeled PI3Kβ on supported lipid bilayers using single molecule TIRF microscopy. This is the first study to reconstitute membrane localization and activation of a class 1A PI3K using multiple signaling inputs that are all membrane-tethered in a physiologically relevant configurations. Previous experiments have relied exclusively on phosphotyrosine peptides (pY) presented in solution to activate PI3Kα, PI3Kβ, or PI3Kδ (*Zhang et al., 2011*; *Dornan et al., 2017*; *Dbouk et al., 2012*). However, pY peptides are derived from the cytoplasmic domains of transmembrane receptors, such as receptor tyrosine kinases (RTKs), which reside in the plasma membrane (*Lemmon and Schlessinger, 2010*). Although pY peptides in solution can disrupt the inhibitory contacts between the regulatory and catalytic subunits of class 1A PI3Ks (*Zhang et al., 2011*; *Yu et al., 1998*), they do not robustly localize PI3Ks to membranes when they exist in solution. When conjugated to SLBs, we find that pY peptides strongly localize PI3Kβ and relieve autoinhibition, while membranes containing either Rac1(GTP) or GβGγ alone are unable to robustly localize PI3Kβ. We observed this prioritization of signaling input interactions over a range of membrane compositions that contained physiologically relevant densities of anionic lipids, such as 20% PS and 2% PI(4,5)P$_2$. Although a small fraction of PI3Kβ may transiently adopt a conformation that is compatible with direct Rac1(GTP) or GβGγ association in the absence of pY, these events are rare and do not represent the most probable pathway for controlling initial PI3Kβ membrane docking. Given the complexity of the cellular plasma membrane lipid composition (*Lorent et al., 2020*), our experimental system uses a relatively simple mixture of lipids that maximized membrane fluidity and minimized surface defects that could promote non-specific molecular interactions. By this approach, our study provides clarity concerning the strength of various protein-protein interactions that regulate PI3Kβ membrane localization and activity.

Based on our single molecule dwell time and diffusion analysis, Dy647-PI3Kβ can cooperatively bind to one doubly phosphorylated peptide derived from the PDGF receptor. Supporting this model, Dy647-PI3Kβ with a mutated nSH2 or cSH2 domain that eliminates pY binding, still displayed

membrane diffusivity indistinguishable from wild-type PI3Kβ. The diffusion coefficient of membrane-bound pY-PI3Kβ complexes also did not significantly change over a broad range of pY membrane surface densities that span three orders of magnitude. Given that diffusivity of peripheral membrane binding proteins is strongly correlated with the valency of membrane interactions (*Ziemba and Falke, 2013*; *Hansen et al., 2022*), we expected to observe a decrease in Dy647-PI3Kβ diffusion with increasing membrane surface densities of pY. Instead, our data suggests that the vast majority of PI3Kβ molecules engage a single pY peptide, rather than binding one tyrosine phosphorylated residue on two separate pY peptides. While no structural studies have shown how exactly the tandem SH2 domains of PI3K (p85α) simultaneously bind to adoubly phosphorylated pY peptide, the interactions likely resemble the mechanism reported for ZAP-70 (Zeta-Chain-Associated Protein Kinase 70). The tandem SH2 domains of ZAP-70 can bind to a doubly phosphorylated chain derived from the TCR with only 11 amino acids spacing between the two tyrosine phosphorylation sites (*Hatada et al., 1995*). In the case of our PDGFR-derived pY peptide that binds p85α, 10 amino acids separate the two tyrosine phosphorylated residues.

Following the engagement of a pY peptide, we find that PI3Kβ can then associate with membrane-anchored Rac1(GTP) or GβGγ. We detected the formation of PI3Kβ-pY-Rac1(GTP) and PI3Kβ-pY-GβGγ complexes based on the following criteria: (1) increase in Dy647-PI3Kβ bulk membrane recruitment, (2) increase in single molecule dwell time, and (3) a decrease in membrane diffusivity. Consistent with Dy647-PI3Kβ having a weak affinity for GβGγ, pY peptides in solution were unable to strongly localize Dy647-PI3Kβ to SLBs containing membrane-anchored GβGγ unless there was 20% PS lipids present. This is in agreement with HDX-MS data showing that the p110β-GβGγ interaction can only be detected using a GβGγ-p85α(icSH2) chimeric fusion or pre-activating PI3Kβ with solution pY (*Dbouk et al., 2012*). Using AlphaFold Multimer (*Evans et al., 2022*; *Jumper et al., 2021*), we created a model that illustrates how the p85α(nSH2) domain is predicted to sterically block GβGγ binding to p110β. This model was validated by comparing the AlphaFold Multimer model to previous reported HDX-MS (*Dbouk et al., 2010*) and X-ray crystallography data (*Zhang et al., 2011*). Further supporting this model, we found that the Dy647-PI3Kβ nSH2(R358A) mutant tethered to membrane-conjugated pY peptide was unable to engage membrane-anchored GβGγ. Membrane targeting of PI3Kβ by pY was required to relieve nSH2 mediated autoinhibition and expose the GβGγ binding site. Recruitment by membrane-tethered pY also reduces the translational and rotational entropy of PI3Kβ, which facilitates PI3Kβ-pY-GβGγ complex formation. We observed a similar mechanism of synergistic PI3Kβ localization on SLBs containing pY and Rac1(GTP). This was consistent with single molecule studies investigating synergistic enhanced localization of PI3Kα in the presence of H-Ras(GTP) and solution pY peptide (*Buckles et al., 2017*). It remains unclear how p85α-mediated inhibition controls the association dynamics between PI3Kβ and Rac1(GTP).

## Mechanism of synergistic activation

Previous characterization of PI3Kβ lipid kinase activity has utilized solution-based assays to measure P(3,4,5)P$_3$ production. These solution-based measurements lack spatial information concerning the mechanism of PI3Kβ membrane recruitment and activation. Our ability to simultaneously visualize PI3Kβ membrane localization and P(3,4,5)P$_3$ production was critical for determining which regulatory factors directly modulate the catalytic efficiency of PI3Kβ. In the case of PI3Kα, the enhanced membrane recruitment model has been used to explain the synergistic activation mediated by pY and Ras(GTP) (*Buckles et al., 2017*). In other words, the PI3Kα-pY-Ras complex is more robustly localized to membranes compared to the PI3Kα-pY and PI3Kα-Ras complexes, which results in a larger total catalytic output for the system. Although the Ras binding domain (RBD) of PI3Kα and PI3Kβ are conserved, these kinases interact with distinct Ras superfamily GTPases (*Fritsch et al., 2013*). Therefore, it's possible that PI3Kα and PI3Kβ display different mechanisms of synergistic activation, which could explain their non-overlapping roles in cell signaling.

Studies of PI3Kβ mouse knock-in mutations in primary macrophages and neutrophils have shown that robust PI3Kβ activation requires coincident activation through the RTK and GPCR signaling pathways (*Houslay et al., 2016*). This response most strongly depends on the ability of PI3Kβ to bind GβGγ and, to a lesser extent, Rac1/Cdc42 (*Houslay et al., 2016*). A similar mechanism of synergistic activation has been reported for PI3Kγ in the presence of H-Ras(GTP) and GβGγ (*Suire and Lécureuil, 2012*; *Rathinaswamy et al., 2021*). Although mutational studies have deciphered the pathways that

drive synergistic of PI3Kγ and PI3Kβ activation in cells, signaling network crosstalk and redundancy limits our mechanistic understanding of how these kinases prioritizes signaling inputs and the exact mechanism for driving $PI(3,4,5)P_3$ production. Based on our single molecule membrane binding experiments, autoinhibited PI3Kβ does not strongly associate with either Rac1(GTP) or GβGγ in the absence of pY peptides. We found that PI3Kβ kinase activity is also relatively insensitive to either Rac1(GTP) or GβGγ alone but can stimulate some PI3Kβ activity when 20% PS lipids is incorporated into the supported lipid bilayers. Importantly, the prioritization of signal input localization strengths was similar across all membrane compositions tested. Previous studies that showed Rho-GTPases (**Fritsch et al., 2013**) and GβGγ (**Katada et al., 1999**; **Dbouk et al., 2012**; **Maier et al., 1999**) can individually activate PI3Kβ used more complex lipid mixtures that incorporate sphingomyelin, cholesterol, and phosphatidylethanolamine to mimic the cellular plasma membrane composition. In the future, a more comprehensive analysis will be required to map the relationship between PI3Kβ activity, membrane localization, and lipid composition.

In our single molecule TIRF experiments, we find that the pY peptide is the only factor that robustly localizes PI3Kβ to supported membranes in an autonomous manner. However, the pY-PI3Kβ complexdisplays weak lipid kinase activity ($k_{cat}$ = 3 $PI(3,4,5)P_3$ lipids/sec per PI3Kβ). This is consistent with cellular measurements showing that RTK activation by insulin (**Knight et al., 2006**), PDGF (**Guillermet-Guibert et al., 2008**), or EGF (**Ciraolo et al., 2008**) show little PI3Kβ dependence for $PI(3,4,5)P_3$ production. Although the dominant role of PI3Kα in controlling $PI(3,4,5)P_3$ production downstream of RTKs can mask the contribution from PI3Kβ in some cell types, these results highlight the need for PI3Kβ to be synergistically activated. When we measured the kinetics of lipid phosphorylation for PI3Kβ-pY-Rac1(GTP) and PI3Kβ-pY-GβGγ complexes, we observed synergistic activation beyond simply enhancing PI3Kβ membrane localization. After accounting for the ~1.8-fold increase in membrane localization between PI3Kβ-pY-Rac1(GTP) and PI3Kβ-pY-Rac1(GDP), we calculated a 4.3-fold increase in $k_{cat}$ (13 $PI(3,4,5)P_3$ lipids/sec per PI3Kβ) that was dependent on engaging Rac1(GTP). Comparing the kinase activity of PI3Kβ-pY and PI3Kβ-pY-GβGγ complexes that are present at the same membrane surface density (~0.2 PI3Kβ/$\mu m^2$) revealed a 22-fold increase in $k_{cat}$ mediated by the GβGγ interaction. Together, these results indicate that PI3Kβ-pY complex association with either Rac1(GTP) or GβGγ allosterically modulates PI3Kβ, making it more catalytically efficient.

## Mechanisms controlling cellular activation of PI3Kβ

Studies of PI3K activation by pY peptides have mostly been performed using peptides derived from the IRS-1 (Insulin Receptor Substrate 1) and the EGFR/PDGF receptors (**Backer et al., 1992**; **Fantl et al., 1992**). As a result, we still have not defined the broad specificity p85α has for tyrosine-phosphorylated peptides. Biochemistry studies indicate that the nSH2 and cSH2 domains of p85α robustly bind pY residues with a methionine in the +3 position (pYXXM) (**Breeze et al., 1996**; **Nolte et al., 1996**; **Backer et al., 1992**; **Fantl et al., 1992**). The p85α subunit is also predicted to interact with the broad repertoire of receptors that contain immunoreceptor tyrosine-based activation motifs (ITAMs) baring the pYXX(L/I) motif (**Reth, 1989**; **Osman et al., 1996**; **Zenner et al., 1996**; **Love and Hayes, 2010**). Based on RNA seq data, human neutrophils express at least six different Fc receptors (FcRs) that all contain phosphorylated ITAMs that can potentially facilitate membrane localization of class 1A PI3Ks (**Rincón et al., 2018**).

A variety of human diseases result from the overexpression of RTKs, especially the epithelial growth factor receptor (EGFR) (**Sauter et al., 1996**). When the cellular plasma membrane contains densities of EGFR greater than 2000 receptors/$\mu m^2$, trans-autophosphorylation, and activation can occur in an EGF-independent manner (**Endres et al., 2013**). Membrane surface densities above the threshold required for spontaneous trans-autophosphorylation of EGFR have been observed in many cancer cells (**Haigler et al., 1978**). In these disease states, PI3K is expected to localize to the plasma membrane in the absence of ligand-induced RTK or GPCR signaling. The slow rate of $PI(3,4,5)P_3$ production we measured for the membrane-tethered pY-PI3Kβ complex suggests that $PI(3,4,5)P_3$ levels are not likely to rise above the global inhibition imposed by lipid phosphatases until synergistic activation of PI3Kβ by RTKs and GPCRs. However, loss of PTEN in some cancers (**Jia et al., 2008**) could produce an elevated level of $PI(3,4,5)P_3$ due to PI3Kβ being constitutively membrane localized via ligand-independent trans-autophosphorylation of RTKs.

## Materials and methods

### Molecular biology

The following genes were used as templates for PCR to clone plasmids used for recombinant protein expression: *PIK3CB* (human 1-1070aa; Uniprot Accession #P42338), *PIK3R1* (human 1-724aa; Uniprot Accession #P27986), *PIK3CG* (human 1-1102aa; Uniprot Accession #P48736), *PIK3R5* (human 1-880aa; Uniprot Accession #Q8WYR1), *RAC1* (human 1-192aa; Uniprot Accession #P63000), *CYTH3*/Grp1 (human 1-400aa; Uniprot Accession #O43739), *BTK* (bovine 1-659aa; Uniprot Accession #Q3ZC95), neutrophil cytosol factor 2 (*NCF2*, human 1-526aa; Uniprot Accession #P19878, referred to as p67/phox), *PREX1* (human 1-1659aa; Uniprot Accession #Q8TCU6), *GNB1/GBB1* (Gβ$_1$, bovine 1-340aa; Uniprot Accession #P62871), *GNG2/GBG2* (Gγ$_2$, bovine 1-71aa; Uniprot Accession #P63212). The following plasmids were purchased as cDNA clones from Horizon (PerkinElmer), formerly known as Open Biosystems and Dharmacon: human *PIK3R1* (clone #30528412, cat #MHS6278-202806334) and human *CYTH3*/Grp1 (clone #4811560, cat #MHS6278-202806616). Genes encoding bovine Gβ$_1$ and Gγ$_2$ were derived from the following plasmids: YFP-Gβ$_1$ (Addgene plasmid # 36397) and YFP-Gγ$_2$ (Addgene plasmid # 36102). These Gβ$_1$ and Gγ$_2$ containing plasmids were kindly provided to Addgene by Narasimhan Gautam (*Saini et al., 2007*). In this study, we used a previously described mutant form of Btk with mutations in the peripheral PI(3,4,5)P$_3$ binding site (R49S/K52S) (*Chung et al., 2019*; *Wang et al., 2015*). The Btk peripheral site mutant was PCR amplified using a plasmid provided by Jean Chung (Colorado State, Fort Collins) that contained the following coding sequence: his6-SUMO-Btk(PH-TH, R49S/K52S)-EGFP. The nSH2 biosensor was derived from human *PIK3R1*. The gene encoding human *PREX1* was provided by Orion Weiner (University of California San Francisco). Refer to supplemental text to see exact peptide sequence of every protein purified in this study. The following mutations were introduced into either the *PIK3CB* (p110β) or *PIK3R1* (p85α) genes using site-directed mutagenesis: p85α nSH2 (R358A, FLVR->FLVA), p85α cSH2 (R649A, FLVR->FLVA), p110β GβGγ mutant (K532D/K533D). For cloning, genes were PCR amplified using AccuPrime *Pfx* master mix (Thermo Fisher, Cat#12344040) and combined with a restriction digested plasmids using Gibson assembly (*Gibson et al., 2009*). Refer to the Appendix Resource Table for a complete list of plasmids used in this study. The complete open reading frame of all vectors used in this study were sequenced to ensure the plasmids lacked deleterious mutations.

### BACMID and baculovirus production

We generated BACMID DNA as previously described (*Hansen et al., 2019*). FASTBac1 plasmids containing our gene of interested were transformed into DH10 Bac cells and plated on LB agar media containing 50 µg/mL kanamycin, 10 µg/mL tetracycline, 7 µg/mL gentamycin, 40 µg/mL X-GAL, and 40 µg/mL IPTG. Plated cells were incubated for 2–3 days at 37 °C before positive clones were isolated based on blue-white colony selection. White colonies were inoculated into 5 mL of TPM containing 50 µg/mL kanamycin, 10 µg/mL tetracycline, 7 µg/mL gentamycin, and grown overnight at 37 °C. To purify the BACMID DNA, bacteria were pelleted by centrifugation and then re-suspended in 300 µL of buffer containing 50 mM Tris [pH 8.0], 10 mM EDTA, 100 µg/mL RNase A. Bacteria were lysed into 300 µL of buffer containing 200 mM NaOH, 1% SDS before neutralization with 300 µL of 4.2 M Guanidine HCl, 0.9 M KOAc [pH 4.8]. Samples we subsequently centrifuged at 23 °C for 10 min at 14,000 × g. Supernatant containing the BACMID DNA was combined with 700 µL 100% isopropanol and spun for 10 min at 14,000 × g. The DNA pellets were washed twice with 70% ethanol (200 µL and 50 µL) and centrifuged. The ethanol was removed by vacuum aspiration and the final DNA pellet was dried in a biosafety hood. Finally, we solubilized the BACMID DNA in 50–100 µL of sterile filtered MilliQwater. A Nanodrop was used to quantify the total DNA concentration. BACMID DNA was stored at –20 °C or used immediately for higher transfection efficiency. Baculovirus was generated as previously described. In brief, we incubated 5–7 µg of BACMID DNA with 4 µL Fugene (Thermo Fisher, Cat# 10362100) in 250 µL of Opti-MEM serum-free media for 30 min at 23 °C. The DNA-Fugene mixture was then added to a Corning six-well plastic dish (Cat# 07-200-80) containing 1 × 10$^6$ *Spodoptera frugiperda* (Sf9) insect cells in 2 mL of ESF 921 Serum-Free Insect Cell Culture media (Expression Systems, Cat# 96–001, Davis, CA.). 4–5 days following the initial transfection, we harvested and centrifuged the viral supernatant (called 'P0'). P0 was used to generate a P1 titer by infecting 7 × 10$^6$ *Sf9* cells plated on a 10 cm tissue culture grade petri dish containing 10 mL of ESF 921 media and 2% Fetal Bovine serum (Seradigm, Cat# 1500–500, Lot# 176B14). After 5 days of transfection, the P1 titer was harvest and

clarified by centrifugation. The P1 titer was expanded by infecting 100 mL Sf9 cell culture grown to a density of $10^6$ cells/mL with 1% vol/vol P1 viral titer in a sterile 250 mL polycarbonate Erlenmeyer flask with a vented cap (Corning, #431144). The P2 titer (viral supernatant) was harvested, centrifuged, and 0.22 µm filtered in 150 mL filter-top bottle (Corning, polyethersulfone (PES), Cat#431153). The P2 viral titer was subsequently used for protein expression in High five cells grown in ESF 921 Serum-Free Insect Cell Culture media (0% FBS) using a final baculovirus concentration of ~2% vol/vol. All our media contained 1 x concentration of Antibiotic-Antimycotic (Gibco/Invitrogen, Cat#15240–062). A PCR-based Mycoplasma Detection kit was used to verify our insect cell lines were free of mycoplasma contaminants (ATCC, Cat #30–1012 K).

## Protein purification

### PI3Kβ and PI3K γ

Genes encoding human his6-TEV-PIK3CB (1-1070aa) and ybbr-PIK3R1 (1-724aa) were cloned into a modified FastBac1 dual expression vector containing tandem polyhedrin (pH) promoters. Genes encoding human PIK3CG (1-1102aa) and TwinStreptTag-his10-TEV-ybbr-PIK3R5 (1-880aa) were expressed and purified using separate FastBac1 vectors under the polyhedrin (pH) promoters as previously described (*Rathinaswamy et al., 2021*). For protein expression, high titer baculovirus was generated by transfecting 1 × $10^6$ *Spodoptera frugiperda* (Sf9) with 0.75–1 µg of BACMID DNA as previously described (*Hansen et al., 2019*). After two rounds of baculovirus amplification and protein test expression, 2 × $10^6$ cells/mL High five cells were infected with 2% vol/vol PI3Kβ (PIK3CB/PIK3R1) or 2% vol/vol PI3Kγ (PIK3CG/PIK3R5) baculovirus and grown at 27 °C in ESF 921 Serum-Free Insect Cell Culture media (Expression Systems, Cat# 96–001) for 48 hrs. High five cells were harvested by centrifugation and washed with 1 x PBS [pH 7.2] and centrifuged again. Final cell pellets were resuspended in an equal volume of 1 x PBS [pH 7.2] buffer containing 10% glycerol and 2 x protease inhibitor cocktail (Sigma, Cat# P5726) before being stored in the –80 °C freezer. For protein purification, frozen cell pellets from 4 liters of cell culture were lysed by Dounce homogenization into buffer containing 50 mM $Na_2HPO_4$ [pH 8.0], 10 mM imidazole, 400 mM NaCl, 5% glycerol, 2 mM PMSF, 5 mM BME, 100 µg/mL DNase, 1 x protease inhibitor cocktail (Sigma, Cat# P5726). Lysate was centrifuged at 35,000 rpm (140,000 × $g$) for 60 min under vacuum in a Beckman centrifuge using a Ti-45 rotor at 4 °C. Lysate was batch bound to 5 mL of HisPur Ni-NTA Superflow Agarose (Thermo Scientific, Cat #25216) resin for 90 min stirring in a beaker at 4 °C. Resin was washed with buffer containing 50 mM $Na_2HPO_4$ [pH 8.0], 30 mM imidazole, 400 mM NaCl, and 5 mM BME. Protein was eluted from Ni-NTA resin with wash buffer containing 500 mM imidazole. The his6-TEV-PIK3CB/ybbr-PIK3R1 complex was then desalted on a G25 Sephadex column in buffer containing 20 mM Tris [pH 8.0], 100 mM NaCl, 1 mM DTT. Peak fractions were pooled and loaded onto a Heparin anion exchange column equilibrated in 20 mM Tris [pH 8.0], 100 mM NaCl, 1 mM DTT buffer. Proteins were resolved over a 10–100% linear gradient (0.1–1 M NaCl) at 2 mL/min flow rate over 20 min. Peak fractions were pooled and supplemented with 10% glycerol, 0.05% CHAPS, and 200 µg/mL his6-TEV(S291V) protease. The his6-TEV-PIK3CB/ybbr-PIK3R1 complex was incubated overnight at 4 °C with TEV protease to cleave off the his6 affinity tag. The TEV protease cleaved PIK3CB/ybbr-PIK3R1 complex was separated on a Superdex 200 size exclusion column (GE Healthcare, Cat# 17-5174-01) equilibrated with 20 mM Tris [pH 8.0], 150 mM NaCl, 10% glycerol, 1 mM TCEP, 0.05% CHAPS. Peak fractions were concentrated in a 50 kDa MWCO Amicon centrifuge tube and snap frozen at a final concentration of 10 µM using liquid nitrogen. This protein is referred to as PI3Kβ throughout the manuscript. The same protocol was followed to purify the various PI3Kβ mutants reported in this study.

### Rac1

The gene encoding human Rac1 were expressed in BL21 (DE3) bacteria as his10-SUMO3-(Gly)$_5$-Rac1 fusion protein. Bacteria were grown at 37 °C in 4 L of Terrific Broth for 2 hrs or until $OD_{600}=0.8$. Cultures were shifted to 18 °C for 1 hour and then induced with 0.1 mM IPTG. Expression was allowed to continue for 20 hr before harvesting. Cells were lysed into 50 mM $Na_2HPO_4$ [pH 8.0], 400 mM NaCl, 0.4 mM BME, 1 mM PMSF, 100 µg/mL DNase using a microfluidizer. Lysate was centrifuged at 16,000 rpm (35,000 × g) for 60 min in a Beckman JA-20 rotor at 4 °C. Lysate was circulated over a 5 mL HiTrap Chelating column (GE Healthcare, Cat# 17-0409-01) loaded with $CoCl_2$. Bound protein was eluted at a flow rate of 4 mL/min into 50 mM $Na_2HPO_4$ [pH 8.0], 400 mM NaCl, 500 mM imidazole.

Peak fractions were pooled and combined with SUMO protease (his6-SenP2) at a final concentration of 50 µg/mL and dialyzed against 4 liters of buffer containing 20 mM Tris [pH 8.0], 250 mM NaCl, 10% Glycerol, 1 mM MgCl$_2$, 0.4 mM BME. Dialysate containing SUMO cleaved protein was recirculated for 2 hrs over a 5 mL HiTrap Chelating column. Flowthrough containing (Gly)$_5$-Rac1 was concentrated in a 5 MWCO Vivaspin 20 before being loaded on a 124 mL Superdex 75 column equilibrated in 20 mM Tris [pH 8.0], 150 mM NaCl, 10% Glycerol, 1 mM TCEP, 1 mM MgCl$_2$. Peak fractions containing (Gly)$_5$-Rac1 were pooled and concentrated to 400–500 µM (~10 mg/mL) and snap frozen with liquid nitrogen and stored at –80 °C.

### Grp1 and nSH2

The gene encoding the Grp1 PH domain derived from human CYTH3 was expressed in BL21 (DE3) bacteria as a his6-MBP-N10-TEV-GGGG-Grp1 fusion protein. The gene encoding the N-terminal SH2 (nSH2, 322-440aa) domain derived from the *PIK3R1* gene was cloned and expressed as a his6-GST-TEV-nSH2 fusion protein. A single cysteine was encoded in the C-terminus of the nSH2 domain and Grp1 to allow for chemical labeling with maleimide dyes. For both recombinant proteins, bacteria were grown at 37 °C in 4 L of Terrific Broth for 2 hrs or until OD$_{600}$=0.8 and then shifted to 18 °C for 1 hr. Cells were then induced to express either the Grp1 or nSH2 fusion by adding 0.1 mM IPTG. Cells were harvested 20 hrs post-induction. Bacteria were lysed into 50 mM Na$_2$HPO$_4$ [pH 8.0], 400 mM NaCl, 0.4 mM BME, 1 mM PMSF, 100 µg/mL DNase using a microfluidizer. Next, lysate was centrifuged at 16,000 rpm (35,000 × g) for 60 min in a Beckman JA-20 rotor at 4 °C. Supernatant was circulated over 5 mL HiTrap chelating column (GE Healthcare, Cat# 17-0409-01) that was pre-incubated with 100 mM CoCl$_2$ for 10 min, wash with MilliQ water, and equilibrated into lysis buffer lacking PMSF and DNase. Clarified cell lysate containing his6-MBP-N10-TEV-GGGG-Grp1 was circulated over the HiTrap column and washed with 20 column volumes of 50 mM Na$_2$HPO$_4$ [pH 8.0], 300 mM NaCl, 0.4 mM BME containing buffer. Protein was eluted with buffer containing 50 mM Na$_2$HPO$_4$ [pH 8.0], 300 mM NaCl, and 500 mM imidazole at a flow rate of 4 mL/min. Peak HiTrap elution fractions were combined with 750 µL of 2 mg/mL TEV protease and dialyzed overnight against 4 L of buffer containing 20 mM Tris [pH 8.0], 200 mM NaCl, and 0.4 mM BME. The next day, we recirculated cleaved proteins over two HiTrap (Co$^{+2}$) columns (2 × 5 mL) that were equilibrated in 50 mM Na$_2$HPO$_4$ [pH 8.0], 300 mM NaCl, and 0.4 mM BME containing buffer for 1 hr. Proteins were concentrated using a 10 kDa MWCO Vivaspin 20 and then loaded on a 124 mL Superdex 75 column equilibrated in in 20 mM Tris [pH 8], 200 mM NaCl, 10% glycerol, and 1 mM TCEP. Proteins were eluted at a flow rate of 1 mL/min. Peak fractions containing Grp1 were pooled and concentrated to 500–600 µM (~8 mg/mL). Peak fractions containing nSH2 were pooled and concentrated to 200–250 µM (~3 mg/mL). Proteins were frozen with liquid nitrogen and stored at –80 °C.

### P-Rex1 (DH-PH) domain

The DH-PH domain of human P-Rex1 was expressed as a fusion protein, his6-MBP-N10-TEV-PRex1(40-405aa), in BL21(DE3) Star bacteria. Bacteria were grown at 37 °C in 2 L of Terrific Broth for 2 hrs or until OD$_{600}$=0.8. Cultures were shifted to 18 °C for 1 hr then induced with 0.1 mM IPTG. Expression was allowed to continue for 20 hrs before harvesting. Cells were lysed into buffer containing 50 mM NaHPO4 [pH 8.0], 400 mM NaCl, 5% glycerol, 1 mM PMSF, 0.4 mM BME, 100 µg/mL DNase using microtip sonication. Cell lysate was clarified by centrifugation at 16,000 rpm (35,000 × g) for 60 min in a Beckman JA-20 rotor at 4 °C. To capture his$_6$-MBP-N10-TEV-PRex1, cell lysate was circulated over a 5 mL HiTrap Chelating column (GE Healthcare, Cat# 17-0409-01) charged CoCl$_2$. The column was washed with 100 mL of 50 mM NaHPO4 [pH 8.0], 400 mM NaCl, 5% glycerol, 0.4 mM BME buffer. Protein was eluted into 15 mL with buffer containing 50 mM NaHPO4 [pH 8.0], 400 mM NaCl, 500 mM imidazole, 5% glycerol, 0.4 mM BME. Peak fractions were pooled and combined with his6-TEV protease and dialyzed against 4 liters of buffer containing 50 mM NaHPO4 [pH 8.0], 400 mM NaCl, 5% glycerol, 0.4 mM BME. The next day, dialysate containing TEV protease cleaved protein was recirculated for 2 hrs over a 5 mL HiTrapchelating column. Flowthrough containing P-Rex1 (40-405aa) was desalted into 20 mM Tris [pH 8.0], 50 mM NaCl, 1 mM DTT using a G25 Sephadex column. Note that some of the protein precipitated during the desalting step. Desalted protein was clarified using centrifugation and 0.22 µm syringe filter. P-Rex1(40-405aa, pI = 8.68) was further purified by cation exchange chromatography (i.e. MonoS) using a 20 mM Tris [pH 8.0], 1 mM DTT, and 0.05 to 1 M

NaCl gradient. P-Rex1(40-405aa) bound eluted broadly in the presence of 100–260 mM NaCl. Pure fractions as determined by SDS-PAGE were pooled, spin concentrated, and loaded onto a 120 mL Superdex 75 column equilibrated in 20 mM Tris [pH 8], 150 mM NaCl, 10% glycerol, 1 mM TCEP. Peak fractions containing P-Rex1(40-405aa) were pooled and concentrated to a concentration of 114 µM, aliquoted, frozen with liquid nitrogen, and stored at –80 °C.

### Btk

The mutant BtkPI(3,4,5)P$_3$ fluorescent biosensor was recombinantly expressed in BL21 Star *E. coli* as a his6-SUMO-Btk(1-171aa PH-TH domain; R49S/K52S)-SNAP fusion. Bacteria were grown at 37 °C in Terrific Broth to an OD$_{600}$=0.8. These cultures were then shifted to 18 °C for 1 hr, induced with 0.1 mM IPTG, and allowed to express protein for 20 hrs at 18 °C before being harvested. Cells were lysed into 50 mM NaPO$_4$ (pH 8.0), 400 mM NaCl, 0.5 mM BME, 10 mM Imidazole, and 5% glycerol. Lysate was then centrifuged at 16,000 rpm (35,172 × g) for 60 min in a Beckman JA-20 rotor chilled to 4 °C. Lysate was circulated over 5 mL HiTrap Chelating column (GE Healthcare, Cat# 17-0409-01) charged with 100 mM CoCl$_2$ for 2 hrs. Bound protein was then eluted with a linear gradient of imidazole (0–500 mM, 8 CV, 40 mL total, 2 mL/min flow rate). Peak fractions were pooled, combined with SUMO protease Ulp1 (50 µg/mL final concentration), and dialyzed against 4 L of buffer containing 20 mM Tris [pH 8.0], 200 mM NaCl, and 0.5 mM BME for 16–18 hrs at 4 °C. SUMO protease cleaved Btk was recirculated for 1 hr over a 5 mL HiTrap Chelating column. Flow-through containing Btk-SNAP was then concentrated in a 5 kDa MWCO Vivaspin 20 before being loaded on a Superdex 75 size-exclusion column equilibrated in 20 mM Tris [pH 8.0], 200 mM NaCl, 10% glycerol, 1 mM TCEP. Peak fractions containing Btk-SNAP were pooled and concentrated to a concentration of 30 µM before snap-freezing with liquid nitrogen and storage at –80 °C. For labeling, Btk-SNAP was combined with a 1.5 x molar excess of SNAP-Surface Alexa488 dye (NEB, Cat# S9129S) and incubated overnight at 4 °C. The next day, Btk-SNAP-AF488 was desalted into buffer containing 20 mM Tris [pH 8.0], 200 mM NaCl, 10% glycerol, 1 mM TCEP using a PD10 column. The protein was then spin concentrated using a Amicon filter and loaded onto a Superdex 75 column to isolate dye free monodispersed Btk-SNAP-AF488. The peak elution was pooled, concentrated, aliquoted, and flash frozen with liquid nitrogen.

### p67/phox

Genes encoding the Rac1(GTP) biosensor, p67/phox, were cloned into a plasmid as his10-TEV-SUMO-p67/phox fusion proteins and expressed in Rosetta2 (DE3) pLysS bacteria. Bacteria were grown in 3 L of Terrific Broth 37 °C for 2 hrs or until OD$_{600}$=0.8 before shifting temperature to 18 °C for 1 hr. Protein expression was induced by adding 50 µM IPTG. Cells expressed overnight for 20 hrs at 18 °C before harvesting. We lysted cells into buffer containing 50 mM Na$_2$HPO$_4$ [pH 8.0], 400 mM NaCl, 0.4 mM BME, 1 mM PMSF, and 100 µg/mL DNase using a microfluidizer. The lysate was centrifuged at 16,000 rpm (35,000 × g) for 60 min in a Beckman JA-20 rotor at 4 °C. Supernatant was then circulated over 5 mL HiTrap Chelating column (GE Healthcare, Cat# 17-0409-01) load with 100 mM CoCl$_2$ for 10 min. The HiTrap column was washed with 20 column volumes (100 mL) of 50 mM Na$_2$HPO$_4$ [pH 8.0], 400 mM NaCl, 10 mM imidazole, and 0.4 mM BME containing buffer. Bound protein was eluted at a flow rate of 4 mL/min with 15–20 mL of 50 mM Na$_2$HPO$_4$ [pH 8.0], 400 mM NaCl, and 500 mM imidazole-containing buffer. Peak fractions were pooled and combined with his6-SenP2 (SUMO protease) at a final concentration of 50 µg/mL and dialyzed against 4 liters of buffer containing 25 mM Tris [pH 8.0], 400 mM NaCl, and 0.4 mM BME. Dialysate containing SUMO cleaved protein was recirculated for 2 hrs over two 5 mL HiTrap Chelating (Co$^{2+}$) columns that were equilibrated in buffer containing 25 mM Tris [pH 8.0], 400 mM NaCl, and 0.4 mM BME. Recirculated protein was concentrated to a volume of 5 mL using a 5 kDa MWCO Vivaspin 20 before loading on a 124 mL Superdex 75 column at a flow rate of 1 mL/min. The column was equilibrated in buffer containing 20 mM HEPES [pH 7], 200 mM NaCl, 10% glycerol, and 1 mM TCEP. Peak fractions off the Superdex 75 column were concentrated in a 5 kDa MWCO Vivaspin 20 to a concentration between 200–500 µM (5–12 mg/mL). Protein was frozen with liquid nitrogen and stored at –80 °C.

## Farnesylated G$_{\beta 1}$Gγ$_2$ and SNAP-G$_{\beta 1}$Gγ$_2$

The native eukaryotic farnesyl Gβ$_1$/Gγ$_2$ and SNAP-Gβ$_1$/Gγ$_2$ complexes were expressed and purified from insect cells as previously described (*Rathinaswamy et al., 2021*; *Kozasa and Gilman, 1995*;

*Dbouk et al., 2012*). The $G\beta_1$ and $G\gamma_2$ genes were cloned into dual expression vectors containing tandem polyhedron promoters. A single baculovirus expressing either $G\beta_1$/his$_6$-TEV-$G\gamma_2$ or SNAP-$G\beta_1$/his6-TEV-$G\gamma_2$ were used to infect 2–4 liters of High Five cells ($2 \times 10^6$ cells/mL) with 2% vol/vol of baculovirus. Cultures were then grown in shaker flasks (120 rpm) for 48 hr at 27 °C before harvesting cells by centrifugation. Insect cells pellets were stored as 10 g pellets in the –80 °C before purification. To isolate farnesylated $G\beta_1$/his$_6$-TEV-$G\gamma_2$ or SNAP-$G\beta_1$/his6-TEV-$G\gamma_2$ complexes, insect cells were lysed by microtip sonication in buffer containing 50 mM HEPES-NaOH [pH 8], 100 mM NaCl, 3 mM MgCl$_2$, 0.1 mM EDTA, 10 µM GDP, 10 mM BME, Sigma PI tablets (Cat #P5726), 1 mM PMSF, DNase (GoldBio, Cat# D-303–1). Homogenized cell lysate was centrifuged for 10 min at $800 \times g$ to remove nuclei and whole cell debris. The supernatant was then centrifuged for 30 min at 4°C in a Beckman Ti45 rotor at $100,000 \times g$. The post-centrifugation pellet containing plasma membranes was resuspended in a buffer containing 50 mM HEPES-NaOH [pH 8], 50 mM NaCl, 3 mM MgCl$_2$, 1% sodium deoxycholate (wt/vol, Sigma D6750), 10 µM GDP (Sigma, cat# G7127), 10 mM BME, and a Sigma Protease Inhibitor tablet (Cat #P5726) to a concentration of 5 mg/mL total protein. A dounce homogenizer was then used to homogenize membranes in the detergent containing buffer. The homogenized solution was then allowed to stir for 1 hr at 4 °C. The solubilized extracted membrane solution was then centrifuged in a Beckman Ti45 rotor $100,000 \times g$ for 45 min at 4 °C. The supernatant containing solubilized $G\beta_1$/his$_6$-TEV-$G\gamma_2$ or SNAP-$G\beta_1$/his6-TEV-$G\gamma_2$ was diluted into buffer containing 20 mM HEPES-NaOH [pH 7.7], 100 mM NaCl, 0.1% $C_{12}E_{10}$ (Polyoxyethylene (10) lauryl ether; Sigma, P9769), 25 mM imidazole, and 2 mM BME.

Soluble membrane extracted $G\beta_1$/his$_6$-TEV-$G\gamma_2$ or SNAP-$G\beta_1$/his6-TEV-$G\gamma_2$ was purified by affinity chromagraphy using HisPur Ni-NTA Superflow Agarose (Thermo Scientific, Cat #25216). After adding Ni-NTA resin to the diluted solubilized extracted membrane solution, the resin and membrane solubilized protein was allowed to stir in a beaker at 4 °C for 2 hrs. The protein-bound resin beads were then packed into a gravity flow column and washed with 20 column volumes of buffer containing 20 mM HEPES-NaOH [pH 7.7], 100 mM NaCl, 0.1% $C_{12}E_{10}$, 20 mM imidazole, and 2 mM BME. To dissociate the bound G alpha subunit from the $G\beta_1$/$G\gamma_2$ complex, the resin was washed with warm buffer (30 °C) containing 20 mM HEPES-NaOH [pH 7.7], 100 mM NaCl, 0.1% $C_{12}E_{10}$, 20 mM imidazole, 2 mM BME, 50 mM MgCl$_2$, 10 µM GDP, 30 µM AlCl$_3$ (J.T. Baker 5–0660), and 10 mM NaF. Finally, the $G\beta_1$/his$_6$-TEV-$G\gamma_2$ or SNAP-$G\beta_1$/his6-TEV-$G\gamma_2$ was eluted from the NiNTA resin with buffer containing 20 mM Tris-HCl (pH 8.0), 25 mM NaCl, 0.1% $C_{12}E_{10}$, 200 mM imidazole, and 2 mM BME. The eluted protein was incubated overnight at 4 °C with TEV protease to cleave off the his6 affinity tag.

The next day, the cleaved protein was desalted on a G25 Sephadex column into buffer containing 20 mM Tris-HCl [pH 8.0], 25 mM NaCl, 8 mM CHAPS, and 2 mM TCEP. Buffer exchanged protein was loaded on an anion exchange chromatography column (i.e. MonoQ) and eluted in the presence of 175-200 mM NaCl. Peak-containing fractions were combined and concentrated using a Millipore Amicon Ultra-4 (10 kDa MWCO) centrifuge filter. Concentrated samples of $G\beta_1$/$G\gamma_2$ or SNAP-$G\beta_1$/$G\gamma_2$, respectively, were loaded on either Superdex 75 or Superdex 200 gel filtration columns equilibrated 20 mM Tris [pH 8.0], 100 mM NaCl, 8 mM CHAPS, and 2 mM TCEP. Peak fractions were combined and concentrated in a Millipore Amicon Ultra-4 (10 kDa MWCO) centrifuge tube. Finally, $G\beta_1$/$G\gamma_2$ and SNAP-$G\beta_1$/$G\gamma_2$ were aliquoted and flash frozen with liquid nitrogen before storing at –80 °C.

## Fluorescent labeling of SNAP-$G\beta_1$/$G\gamma_2$

To fluorescently label SNAP-$G\beta_1$/$G\gamma_2$, the protein complex was combined with 1.5 x molar excess of SNAP-Surface Alexa488 dye (NEB, Cat# S9129S). SNAP dye labeling was performed in buffer containing 20 mM Tris [pH 8.0], 100 mM NaCl, 8 mM CHAPS, and 2 mM TCEP overnight at 4 °C. Labeled protein was then separated from free Alexa488-SNAP surface dye using a 10 kDa MWCO Amicon spin concentrator followed by size exclusion chromatography (Superdex 75 10/300 GL) in buffer containing 20 mM Tris [pH 8.0], 100 mM NaCl, 8 mM CHAPS, 1 mM TCEP. Peak SEC fractions containing Alexa488-SNAP-$G\beta_1$/$G\gamma_2$ were pooled and centrifuged in a 10 kDa MWCO Amicon spin concentrator to reach a final concentration of 15–20 µM before snap freezing in liquid nitrogen and storiage in the –80 °C. To calculate the SNAP dye labeling efficiency, we determined that Alexa488 contributes 11% of the peak $A_{494}$ signal to the measured $A_{280}$. Note that Alexa488 non-intuitively has a peak absorbance at 494 nm. The final concentration of Alexa488-SNAP-$G\beta_1$/$G\gamma_2$ was calculated after

adjusting the $A_{280}$ (i.e. $A_{280(protein)}$ = $A_{280(observed)}$ – $A_{494(dye)}$*0.11) and using the following extinction coefficients: $\varepsilon_{280(SNAP-G\beta1/G\gamma2)}$=78,380 $M^{-1} \times cm^{-1}$, $\varepsilon_{494(Alexa488)}$ = 71,000 $M^{-1} \times cm^{-1}$.

## Fluorescent labeling of PI3K using Sfp transferase

As previously described (*Rathinaswamy et al., 2021*), Dyomics647-CoA was created by incubating a mixture of 15 mM Dyomics647 maleimide (Dyomics, Cat #647P1-03) in DMSO with 10 mM CoA (Sigma, #C3019, MW = 785.33 g/mole) overnight at 23 °C. To quench excess unreacted Dyomics647 maleimide, 5 mM DTT was added to the reaction mixture. Fluorescent labeling of purified PIK3CB/ybbr-PIK3R1 (referred to as PI3Kβ or p110β-p85α in manuscript) was achieved by mixing Dyomics647-CoA and Sfp-his$_6$. The ybbrR13 motif fused to PIK3R1 contained the following peptide sequence: DSLEFI-ASKLA (*Yin et al., 2006*). In a total reaction volume of 2 mL, a combination of 5 μM PI3Kβ, 4 μM Sfp-his$_6$, and 10 μM DY647-CoA was mixed in buffer containing 20 mM Tris [pH 8], 150 mM NaCl, 10 mM MgCl$_2$, 10% Glycerol, 1 mM TCEP, and 0.05% CHAPS. The ybbr labeling reaction was allowed to proceed for 4 hrs on ice. Excess Dyomics647-CoA was removed using a gravity flow PD-10 desalting column. Fluorescently labeled Dy647-PI3Kβ was spin concentrated in a 50 kDa MWCO Amicon centrifuge tube before loading on a Superdex 200 gel filtration column equilibrated in 20 mM Tris [pH 8], 150 mM NaCl, 10% glycerol, 1 mM TCEP, and 0.05% CHAPS (GoldBio, Cat# C-080–100). Peak fractions were pooled and concentrated to 5–10 μM before being aliquoted and flash frozen with liquid nitrogen. The final Dy647-PI3Kβ was stored at –80 °C.

## Preparation of supported lipid bilayers

Small unilamellar vesicles (SUVs) were generated using the following lipids: 1,2-dioleoyl-sn-glycero-3-phosphocholine (18:1 DOPC, Avanti #850375C), 1,2-dioleoyl-*sn*-glycero-3-phospho-L-serine (18:1 DOPS, Avanti #840035C), 1,2-dioleoyl-sn-glycero-3-phosphoethanolamine (18:1 (Δ9-Cis) DOPE, Avanti # 850725C), L-α-phosphatidylinositol-4,5-bisphosphate (Brain PI(4,5)P$_2$, Avanti #840046X), synthetic phosphatidylinositol 4,5-bisphosphate 18:0/20:4 (PI(4,5)P$_2$, Echelon, P-4524), and 1,2-dioleoyl-sn-glycero-3-phosphoethanolamine-N-[4-(p-maleimidomethyl)cyclohexane-carboxamide] (18:1 MCC-PE, Avanti #780201C). All lipid mixtures are based on percentages that are equivalent to molar fractions. Each lipid mixture contained 2 μmoles of total lipids combined with 2 mL of chloroform in a 35 mL glass round bottom flask containing. This mixture was dried to a thin film using rotary evaporation in a glass round-bottom flask incubated in a 42 °C water bath. Following the chloroform rotoevaporation, the lipid-containing flask was flushed with nitrogen gas or placed in a vacuum desiccator for at least 30 min. A final solution concentration of 1 mM lipids was achieved by resuspending the dried film in 2 mL of 1 x PBS [pH 7.2]. Extrusion of the 1 mM lipid mixture through a 0.03 μm pore size 19 mm polycarbonate membrane (Avanti #610002) with filter supports (Avanti #610014) on both sides of the PC membrane was used to generate 30–50 nm SUVs. Prior to creating SLBs, coverglass (25 x 75 mm, IBIDI, cat #10812) was cleaned in a warm (60–70°C) solution of 2% Hellmanex III (Fisher, Cat#14-385-864) in a glass coplin jar. After 30 min in warm 2% Hellmanex III , coverglass was rinsed at least 7 times with MilliQ water. The cleaned glass was then etched with Piranha solution (1:3, hydrogen peroxide:sulfuric acid) for 5–10 min. Etched glass was then rinsed with a copious volume of MilliQ water and then stored in the glass coplin jar containing MilliQ water. To adhere the Piranha etched coverglass to a 6-well sticky-side chamber (IBIDI, Cat# 80608), individual coverglass were rapidly dried with a stream of nitrogen gas. SLBs were created by flowing 100-150 μL of SUVs with a total lipid concentration of 0.25 mM in 1 x PBS [pH 7.2] into the IBIDI chamber. Following 30 min of incubation, supported membranes were washed with 4 mL of 1 x PBS [pH 7.2] to remove non-absorbed SUVs. To block the membrane defects, a 10 mg/mL beta casein solution (ThermoFisherSci, Cat# 37528) was clarified by centrifugation at 4 °C for 30 min at 21,370 x *g* and then passed through a 0.22 μm PES syringe filter (Foxx Life Sciences, Cat#381–2116-OEM). Membrane defects were blocked by incubation with 1 mg/mL beta casein for 5-10 minutes.

When reconstituting amphiphilic molecules (i.e. lipids) in aqueous solution a variety of structures can form based on the lipid composition, including micelles, inverted micelles, and planar bilayers (*Kulkarni, 2019*). The organization of these membrane structures is related to the molecular packing parameter of the individual phospholipids (*Israelachvili et al., 1976*). The packing parameter ($P = v/\left(al_c\right)$) depends on the volume of the hydrocarbon (*v*), area of the lipid head group (*a*), and the lipid tail length (*l$_c$*). When generating supported lipid bilayers on a flat two-dimensional glass

surface, this study sought to create fluid lamellar membranes. Using phosphatidylcholine (PC) lipids was ideal for making supported lipid bilayers because they have a packing parameter of ~1 (*Costigan et al., 2000*). In other words, PC lipids are cylindrical like a paper towel roll. In contrast, cholesterol and phosphatidylethanolamine (PE) lipids have packing parameters of 1.22 and 1.11, respectively (*Angelov et al., 1999*; *Carnie et al., 1979*). This gives cholesterol and PE lipids an inverted truncated cone shape, which prefers to adopt a non-lamellar phase structure when present at high concentrations. Due to the intrinsic negative curvature of PE lipids, they can spontaneously form inverted micelles (i.e. hexagonal II phase) in aqueous solution when they are the predominant lipid species (*Israelachvili et al., 1980*; *Kobierski et al., 2022*; *Wnętrzak et al., 2013*). In this study, incorporation of PE lipids dramatically reduced the protein-maleimide coupling efficiency, increased membrane defects, and resulted in a larger fraction of surface immobilized Dy647-PI3Kβ. This could be related to the intrinsic negative curvature of PE membranes. However, further investigation is needed to decipher these issues.

## Protein conjugation of maleimide lipid

After blocking SLBs with beta-casein, membranes were washed with 2 mL of 1 x PBS and stored at room temperature for up to 2 hrs before mounting on the TIRF microscope. Prior to single molecule imaging experiments, supported membranes were washed into TIRF imaging buffer. Supported membranes containing MCC-PE lipids were used to covalently couple either Rac1(GDP) or the phosphotyrosine peptide (pY). For the pY peptide experiments we used a doubly phosphorylated peptide derived from the mouse platelet derived growth factor receptor (PDGFR) with the following sequence: CSDGG(pY)MDMSKDESID(pY)VPMLDMKGDIKYADIE (33aa). The Alexa488-pY contained the same sequence with the dye conjugated to the C-terminus of the peptide. The peptides were synthesized to >95% purity by ELIM Biopharmaceuticals (Hayward, CA.). For these SLBs, 100 μL of 30 μM Rac1 was diluted in a 1 x PBS [pH 7.2] and 0.1 mM TCEP buffer was added to the IBIDI chamber and incubated for 2 hrs at 23 °C. Importantly, the addition of 0.1 mM TCEP significantly increased the coupling efficiency. SLBs with MCC-PE lipids were then washed with 2 mL of 1 x PBS [pH 7.2] containing 5 mM beta-mercaptoethanol (BME) and incubated for 15 min to neutralize the unreacted maleimide headgroups. SLBs were washed with 1 mL of 1 x PBS, followed by 1 mL of kinase buffer before starting smTIRF-M experiments.

## Nucleotide exchange of Rac1

Membrane conjugated Rac1(GDP) was converted to Rac1(GTP) using either chemical activation (i.e. EDTA/GTP/MgCl$_2$) or the guanine nucleotide exchange factor (GEF), P-Rex1 (DH-PH domain). Chemical activation was accomplished by washing supported membranes containing maleimide linked Rac1(GDP) with 1 x PBS [pH 7.2] containing 1 mM EDTA and 1 mM GTP. Following a 15 min incubation to exchange GDP for GTP, chambers were washed 1 x PBS [pH 7.2] containing 1 mM MgCl$_2$ and 50 μM GTP. A complementary approach that utilizes GEF-mediated activation of Rac1 was achieved by flowing 50 nM P-Rex1 (DH-PH domain) over Rac1(GDP) conjugated membranes (*Figure 1C*). Nucleotide exchange was carried out in buffer containing 1 x PBS, 1 mM MgCl$_2$, 50 μM GTP. Both methods of activation yielded the same density of active Rac1(GTP). Nucleotide exchange of membrane-tethered Rac1 was assessed by visualizing the localization of the Cy3-p67/phox Rac1(GTP) sensor using TIRF-M.

## Single molecule TIRF microscopy

All supported membrane TIRF-M experiments were performed in buffer containing 20 mM HEPES [pH 7.0], 150 mM NaCl, 1 mM ATP, 5 mM MgCl$_2$, 0.5 mM EGTA, 20 mM glucose, 200 μg/mL beta casein (Thermo Scientific, Cat# 37528), 20 mM BME, 320 μg/mL glucose oxidase (Biophoretics, Cat #B01357.02 *Aspergillus niger*), 50 μg/mL catalase (Sigma, #C40-100MG Bovine Liver), and 2 mM Trolox (Cayman Chemicals, Cat#10011659). Perishable reagents (i.e. glucose oxidase, catalase, and Trolox) were added 5–10 min before image acquisition.

## Microscope hardware and imaging acquisition

Single molecule imaging experiments were performed at room temperature (23 °C) using an inverted Nikon Ti2 microscope using a 100 x oil immersion Nikon TIRF objective (1.49 NA). The x-axis and y-axis positions were controlled using a Nikon motorized stage, joystick, and Nikon's NIS element software.

Fluorescently labeled proteins were excited with one of three diode lasers: a 488 nm, a 561 nm, or 642 nm (OBIS laser diode, Coherent Inc, Santa Clara, CA). The lasers were controlled with a Vortran laser launch and acousto-optic tuneable filters (AOTF) control. Excitation and emission light was transmitted through a multi-bandpass quad filter cube (C-TIRF ULTRA HI S/N QUAD 405/488/561/642; Semrock) containing a dichroic mirror. The laser power measured through the objective for single particle visualized was 1–3 mW. Fluorescence emission was captured on an iXon Life 897 EMCCD camera (Andor Technology Ltd., UK) after passing through one of the following 25 mm a Nikon Ti2 emission filters mounted in a Nikon emission filter wheel: ET525/50M, ET600/50M, and ET700/75M (Semrock).

## Kinetic measurements of PI(3,4,5)P$_3$ lipid production

The production of PI(3,4,5)P$_3$ was measured on SLBs in the presence of 20 nM Btk-SNAP-AF488 and visualized using TIRF microscopy. The reaction buffer for lipid phosphorylation reactions contained 20 mM HEPES (pH 7.0), 150 mM NaCl, 5 mM MgCl$_2$, 1 mM ATP, 0.1 mM GTP, 0.5 mM EGTA, 20 mM glucose, 200 µg/mL beta-casein (Thermo Scientific, Cat#37528), 20 mM BME, 320 µg/mL glucose oxidase (Serva, #22780.01 *Aspergillus niger*), 50 µg/mL catalase (Sigma, #C40-100MG Bovine Liver), and 2 mM Trolox (Cayman Chemicals, Cat#10011659). Approximately 5–10 min before image acquisition, chemicals and enzymes needed for the oxygen scavenging system were added to the TIRF imaging buffer. For experiments where inactive GTPases were coupled to membranes, 0.1 mM GTP was replaced with 0.1 mM GDP. For ATP spike-in experiments, 1 mM ATP was omitted from the TIRF imaging buffer and then added as a 100 mM ATP solution using the volume needed to reach a final concentration of 1 mM ATP.

The concentration of the Btk lipid sensor used for the kinetic assays does not interfere with the kinase activity. By comparing the membrane surface intensity of Btk-SNAP-AF488 measured by TIRF microscopy in the presence of both 20 nM and near-saturating micromolar concentrations, we estimate that <0.1% of the PIP lipids are bound to a lipid sensor at any point during the kinetic experiments. Assuming an average footprint of 0.72 nm$^2$ for phosphatidylcholine (*Carnie et al., 1979*; *Hansen et al., 2019*), supported membranes that contained an initial concentrations of 2% PI(4,5)P$_2$ had a membrane surface density of $2.8 \times 10^4$ PI(4,5)P$_2$ lipids/µm$^2$ at the end of the kinase reaction. The plateau fluorescence intensity of the Btk-SNAP-AF488 sensor was considered to be equivalent to the production of 2% PI(3,4,5)P$_3$ when membranes contained an initial concentration of 2% PI(4,5)P$_2$. The bulk membrane intensity of Btk-SNAP-AF488 was normalized from 0 to 1, and multiplied by the total density of PI(3,4,5)P$_3$ lipids to generate kinetic traces that report the kinetics of PI(3,4,5)P$_3$ production.

When performing these lipid kinase assays, it is critical to simultaneously visualize the localization of Btk-SNAP-AF488 and Dy647-PI3Kβ. Poor quality supported membranes can artificially enhance PI3Kβ activity due to non-specific surface absorption of the kinase. When experiments displayed immobilized Dy647-PI3Kβ molecules, the data was omitted from analysis and experiments were repeated.

## Surface density calibration

The density of membrane-tethered proteins attached to supported lipid bilayers was determined by coupling a defined ratio of either fluorescently labeled Cy3-Rac1 (1:10,000) or Alexa488-pY (1:30,000) in the presence of either 30 µM Rac1 or 10 µM pY, respectively. The membrane surface density of GβGγ was quantified at equilibrium using a combination of AF488-SNAP-GβGγ (bulk signal) and dilute AF555-SNAP-GβGγ (0.0025%; 1:40,000), which allowed fluorescent proteins to be resolved and the single molecule density quantified. Single molecule densities of fluorescently labeled pY, Rac1, and GβGγ were estimated using the ImageJ/Fiji Trackmake Plugin. The total surface density was calculated based on the dilution factor in the presence of dark unlabeled protein (e.g. Rac1(GTP), pY, or SNAP-GβGγ).

## Alphafold2 multimer modeling

AlphaFold2 multimer modeling was used through the Mmseqs2 notebook of ColabFold (*Mirdita et al., 2022*) to make structural predictions of PI3Kβ (p110β/p85α) bound to Gβγ. The pLDDT confidence values consistently scored above 90% for all models, with the predicted aligned error and pLDDT scores for all models are shown in *Figure 3—figure supplement 2*.

## Single particle tracking

Single fluorescent Dy647-PI3Kβ complexes bound to supported lipid bilayers were identified and tracked using the ImageJ/Fiji TrackMate plugin (*Jaqaman et al., 2008*). Data was loaded into ImageJ/Fiji as.nd2 files. Using the LoG detector, fluorescent particles were identified based on their size (~6 pixel diameter), brightness, and signal-to-noise ratio. The LAP tracker was used to generate trajectories that followed particle displacement as a function of time. Particle trajectories were then filtered based on Track Start (remove particles at start of movie), Track End (remove particles at end of movie), Duration (particles track ≥2 frames), Track displacement, and X - Y location (removed particles near the edge of the movie). The output files from TrackMate were then analyzed using Prism 9 graphing software to calculate the dwell times. To calculate the dwell times of membrane-bound proteins we generated cumulative distribution frequency (CDF) plots with the bin size set to image acquisition frame interval (e.g. 52 ms). The $\log_{10}$(1-CDF) was plotted as a function dwell time and fit to a single or double exponential curve. For the double exponential curve fits, the alpha value is the percentage of the fast-dissociating molecules characterized by the time constant, $\tau_1$. A typical data set contained dwell times measured for n≥1000 trajectories repeated as n=3 technical replicates.

Single exponential curve fit:

$$f(x) = e^{(-x/\tau)}$$

Two exponential curve fit:

$$f(x) = \alpha * e^{(-x/\tau 1)} + (1 - \alpha) * e^{(-x/\tau 2)}$$

To calculate the diffusion coefficient ($\mu m^2$/s), the probability density was plotted (i.e. frequency divided by bin size of 0.01 μm) versus step size (μm). The step size distribution was fit to the following models:

Single species model:

$$f(r) = \frac{r}{2D\tau} e^{-\left(\frac{r^2}{4D\tau}\right)}$$

Two species model:

$$f(r) = \alpha \frac{r}{2D_1\tau} e^{-\left(\frac{r^2}{4D_1\tau}\right)} + (1 - \alpha) \frac{r}{2D_2\tau} e^{-\left(\frac{r^2}{4D_2\tau}\right)}$$

## Image processing, statistics, and data analysis

Image analysis was performed on ImageJ/Fiji and MatLab. Curve fitting was performed using Prism 9 GraphPad. The X-fold change in dwell time reported in the main text was calculated by comparing the mean single particle dwell time for different experimental conditions. Note that this is different from directly comparing the calculated dwell time (or exponential decay time constant, $\tau_1$). The X% reduction in diffusion or mobility we report in the main text was calculated by comparing the mean single particle displacement (or step size) measured under different experimental conditions.

## Acknowledgements

We thank John Burke (University of Victoria) for his assistance in generating the AlphaFold2 Multimer model of PI3Kβ bound to GβGγ. We thank Grace Waddell (University of Oregon) for the preliminary characterization of PI3Kβ and Colin Hawkinson (University of Oregon) for assistance with protein purification. We thank Jean Chung (Colorado State, Fort Collins) and Orion Weiner (University of California at San Francisco) for plasmids encoding Btk and P-Rex1 plasmids, respectively. Research was supported by the University of Oregon Start-up funds (SDH), National Science Foundation CAREER Award (SDH, MCB-2048060), Molecular Biology and Biophysics Training Program (BRD, NEW, NIH T32 GM007759), and the Summer Program for Undergraduate Research (SPUR) at the University of

Oregon (GMB). The content is solely the responsibility of the authors and does not necessarily represent the official views of the National Science Foundation.

## Additional information

### Funding

| Funder | Grant reference number | Author |
|---|---|---|
| National Science Foundation | MCB-2048060 | Scott D Hansen |
| National Institute of General Medical Sciences | T32 GM007759 | Benjamin R Duewell Naomi E Wilson |

The funders had no role in study design, data collection and interpretation, or the decision to submit the work for publication.

### Author contributions

Benjamin R Duewell, Naomi E Wilson, Conceptualization, Resources, Data curation, Formal analysis, Validation, Investigation, Visualization, Methodology, Writing – review and editing; Gabriela M Bailey, Sarah E Peabody, Resources, Investigation, Writing – review and editing; Scott D Hansen, Conceptualization, Resources, Data curation, Formal analysis, Supervision, Funding acquisition, Validation, Investigation, Visualization, Methodology, Writing – original draft, Project administration, Writing – review and editing

### Author ORCIDs

Scott D Hansen ⓘ https://orcid.org/0000-0001-7005-6200

Reviewer #1 (Public review): https://doi.org/10.7554/eLife.88991.3.sa1
Reviewer #2 (Public review): https://doi.org/10.7554/eLife.88991.3.sa2
Author response https://doi.org/10.7554/eLife.88991.3.sa3

## Additional files

### Supplementary files

• MDAR checklist

### Data availability

All data generated or analyzed during this study are included in the manuscript and supporting files; source data files have been provided for all main text and figure supplements.

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

## Appendix 1

**Appendix 1—key resources table**

| Reagent type (species) or resource | Designation | Source or reference | Identifiers | Additional information |
|---|---|---|---|---|
| Chemical compound, drug | AccuPrime Pfx master mix | Thermo Fisher | 12344040 | |
| Chemical compound, drug | Pfu Ultra High-Fidelity DNA Polymerase | Agilent | 600380 | |
| Chemical compound, drug | Fugene transfection reagent | Thermo Fisher | 10362100 | |
| Chemical compound, drug | SIGMAFAST Protease Inhibitor Cocktail Tablets, EDTA-Free | Sigma | P8830 | |
| Chemical compound, drug | Beta-mercaptoethanol | Sigma | M3148-100ML | |
| Chemical compound, drug | PMSF | Sigma | P7626 | |
| Chemical compound, drug | Guanosine 5'-diphosphate (GDP) sodium salt hydrate | Sigma | G7127-100MG | |
| Chemical compound, drug | Guanosine 5'-triphosphate (GTP) sodium salt hydrate | Sigma | G8877-250MG | |
| Chemical compound, drug | Sodium deoxycholate | Sigma | D6750 | |
| Chemical compound, drug | $C_{12}E_{10}$ (Polyoxyethylene (10) lauryl ether) | Sigma | P9769 | |
| Chemical compound, drug | CHAPS | Thermo Fisher | J6735909 | |
| Chemical compound, drug | 1,2-dioleoyl-sn-glycero-3-phosphocholine (18:1 DOPC) | Avanti | 850375 C | |
| Chemical compound, drug | 1,2-dioleoyl-sn-glycero-3-phospho-L-serine (18:1 DOPS) | Avanti | 840035 C | |
| Chemical compound, drug | 1,2-dioleoyl-sn-glycero-3-phosphoethanolamine (18:1 Δ9-Cis) DOPE | Avanti | 850725 C | |
| Chemical compound, drug | L-α-phosphatidylinositol-4,5-bisphosphate (Brain PI(4,5)$P_2$) | Avanti | 840046 X | |
| Chemical compound, drug | phosphatidylinositol 4,5-bisphosphate 18:0/20:4 (PI(4,5)$P_2$) | Echelon | P-4524 | |
| Chemical compound, drug | 1,2-dioleoyl-sn-glycero-3-phosphoethanolamine-N-[4-(p-maleimidomethyl) cyclohexane-carboxamide] (18:1 MCC-PE) | Avanti | 780201 C | |
| Chemical compound, drug | 10 x PBS [pH 7.4] | Corning | 46–013 CM | |
| Chemical compound, drug | Trolox | Cayman Chemicals | 10011659 | |
| Chemical compound, drug | Dyomics 647 maleimide | Dyomics | 647 P1-03 | |
| Chemical compound, drug | Cy3 maleimide | Cytiva | PA23031 | |

*Appendix 1 Continued on next page*

*Appendix 1 Continued*

| Reagent type (species) or resource | Designation | Source or reference | Identifiers | Additional information |
|---|---|---|---|---|
| Chemical compound, drug | Cy5 maleimide | Cytiva | PA15131 | |
| Chemical compound, drug | Cy3 Mono NHS Ester | Cytiva | PA13101 | |
| Chemical compound, drug | SNAP-Surface Alexa Fluor 488 | NEB | S9129S | |
| Chemical compound, drug | SNAP-Surface Alexa Fluor 546 | NEB | S9132S | |
| Chemical compound, drug | SNAP-Surface Alexa Fluor 647 | NEB | S9136S | |
| Chemical compound, drug | Coenzyme A | Sigma | C3019 | |
| Chemical compound, drug | Sulfuric acid | Sigma | 58105–2.5 L-PC | |
| Peptide, recombinant protein | glucose oxidase (Aspergillus niger, 225 U/mg) | Biophoretics | B01357.02 | |
| Peptide, recombinant protein | catalase (bovine liver) | Sigma | C40-100MG | |
| Peptide, recombinant protein | 10 mg/mL beta casein solution | ThermoFisher | 37528 | |
| Peptide, recombinant protein | LPETGG | ELIM Biopharm | custom peptide synthesis | >95% purity |
| Peptide, recombinant protein | CSDGG(pY)MDMSKDESID(pY)VPMLDMKGDIKYADIE | ELIM Biopharm | custom peptide synthesis | >95% purity |
| Peptide, recombinant protein | CSDGG(pY)MDMSKDESID(pY)VPMLDMKGDIKYADIE-Alexa488 | ELIM Biopharm | custom peptide synthesis | >95% purity |
| Sequence-based reagent | CCTTTTTGGTAgcaGATGCGTCTACTAAAATGCATGGTG | Integrated DNA Technologies (IDT) | FW_ ybbr-PIK3R1 (R358A) | |
| Sequence-based reagent | GTAGACGCATCtgcTACCAAAAAGGTCCCGTCTGCTGTATC | IDT | RV_ ybbr-PIK3R1 (R358A) | |
| Sequence-based reagent | CTTTTCTTGTCgcgGAGAGCAGTAAACAGGGCTGC | IDT | FW_ ybbr-PIK3R1 (R358A) | |
| Sequence-based reagent | GTTTACTGCTCTCcgcGACAAGAAAAGTGCCATCTCGCTTC | IDT | RV_ ybbr-PIK3R1 (R358A) | |
| Sequence-based reagent | CAAGTCGAGGTGGAgatgacTTTCTTCCTGTATTGAAAGAAATCTTGG | IDT | FW_ his6-TEV-PIK3CB(K532D,K533D) | |
| Sequence-based reagent | CAATACAGGAAGAAAgtcatcTCCACCTCGACTTGACACATTAGCAC | IDT | RV_ his6-TEV-PIK3CB(K532D,K533D) | |
| Recombinant DNA reagent | his10-SUMO3-GGGGG-Rac1(1-192aa) | This paper | pSH752 | Bacterial protein expression plasmid |

*Appendix 1 Continued on next page*

*Appendix 1 Continued*

| Reagent type (species) or resource | Designation | Source or reference | Identifiers | Additional information |
|---|---|---|---|---|
| Recombinant DNA reagent | his6-MBP-N10-TEV-GGGGG-P-Rex1 (40-405aa, DH-PH) | This paper | pSH658 | Bacterial protein expression plasmid |
| Recombinant DNA reagent | his10-SUMO3-p67/phox TRP (Rac1-GTP sensor) | This paper | pSH823 | Bacterial protein expression plasmid |
| Recombinant DNA reagent | his6-GST-TEV-nSH2 PIK3R1(322-440aa)-Cys | This paper | pSH615 | Bacterial protein expression plasmid |
| Recombinant DNA reagent | his6-SUMO-Btk(PH-TH,R49S/K52S)-SNAP | This paper | pSH1313 | Bacterial protein expression plasmid |
| Recombinant DNA reagent | his6-MBP-N10-TEV-GGGGG-Grp1 (261-387aa) | This paper | pSH558 | Bacterial protein expression plasmid |
| Recombinant DNA reagent | his6-TEV-SenP2 (368-589aa, SUMO protease) | This paper | pSH653 | Bacterial protein expression plasmid |
| Recombinant DNA reagent | his6-TEV-PIK3CB (1-1070aa) | This paper | pSH541 | Baculovirus expression plasmid |
| Recombinant DNA reagent | ybbr-PIK3R1 (1-724aa) | This paper | pSH743 | Baculovirus expression plasmid |
| recombinant DNA reagent | ybbr-PIK3R1 (FVLR->FVLA, R358A) | This paper | pSH1045 | Baculovirus expression plasmid |
| Recombinant DNA reagent | ybbr-PIK3R1 (FVLR->FVLA, R358A) | This paper | pSH1046 | Baculovirus expression plasmid |
| Recombinant DNA reagent | his6-TEV-PIK3CB (1-1070aa; K532D,K533D) | This paper | pSH1094 | Baculovirus expression plasmid |
| Recombinant DNA reagent | his6-Gg2, Gb1 (DUAL FastBac) | PMID:34452907 | pSH414 | Baculovirus expression plasmid |
| Recombinant DNA reagent | his6-Gg2, SNAP-Gb1 (DUAL FastBac) | PMID:34452907 | pSH651 | Baculovirus expression plasmid |
| Recombinant DNA reagent | PIK3CG(p110g), TwinStrept-his10-TEV-ybbr-PIK3R5(p101) | PMID:36842083 | HP29 | Baculovirus expression plasmid |
| Software, algorithm | GraphPad Prism 9 | GraphPad | https://www.graphpad.com | |
| Software, algorithm | Chimera | UCSF | https://www.rbvi.ucsf.edu/chimera/ | |
| Software, algorithm | ImageJ/Fiji | ImageJ | https://imagej.net/software/fiji/ | |
| Software, algorithm | Nikon NIS elements | Nikon | https://www.microscope.healthcare.nikon.com/products/software/nis-elements | |

*Appendix 1 Continued on next page*

*Appendix 1 Continued*

| Reagent type (species) or resource | Designation | Source or reference | Identifiers | Additional information |
|---|---|---|---|---|
| Cell line(Spodoptera frugiperda) | Sf9 insect cells | Expression Systems | 94–001 S | |
| Cell line (Trichoplusiani) | High five insect cells | UC Berkeley Barker Hall Tissue Culture Facility | High five insect cells | |
| Other | ESF 921 Serum-Free Insect Cell Culture media | Expression Systems | 96-001-01 | Media for insect cell culture |
| Other | Fetal Bovine serum | Seradigm | 1500–500 | Media for insect cell culture |
| Other | Hellmanex III cleaning solution | Fisher | 14-385-864 | Reagent for cleaning coverslips prior to Pirahna etching |

