## [Editor Report · eLife assessment]

The manuscript describes the synergy among PI3Kβ activators, providing **compelling** results concerning the mechanism of their activation. The particular strengths of the work arise to a great extend from the reconstitution system better mimicking the natural environment of the plasma membrane than previous setups have. The study will be a **landmark** contribution to the signaling field.

---

## [Referee Report · Reviewer #1 (Public review)]

The manuscript aims to provide mechanistic insight into the activation of PI3Kβ by its known regulators tyrosine phosphorylated peptides, GTP-loaded Rac1 and G-protein beta-gamma subunits. To achieve this the authors have used supported lipid bilayers, engineered recombinant peptides and proteins (often tagged with fluorophores) and TIRF microscopy to enable bulk (averages of many molecules) and single molecule quantitation. The great strength of this approach is the precision and clarity of mechanistic insight. Although the study does not use "in transfecto" or in vivo models the experiments are performed using "physiologically-based" conditions and provide a powerful insight into core regulatory principles that will be relevant in vivo.

The results are beautiful, high quality, well controlled and internally consistent (and with other published work that overlaps on some points) and as a result are compelling. The primary conclusion is that the primary regulator of PI3Kβ are tyrosine phosphorylated peptides (and by inference tyrosine phsophorylated receptors/adaptors) and that the other activators can synergise with that input but have relatively weak impacts on their own.

Although the methodology is not easily imported, for reasons of both cost and the experience needed to execute them well, the results have broad importance for the field and reverse an impression that had built in large parts of the broader signalling and PI3K communities that all of the inputs to PI3Kβ were relatively equivalent, however, these conclusions were based on "in cell" or in vivo studies that were very difficult to interpret clearly.

---

## [Referee Report · Reviewer #2 (Public review)]

The manuscript of Duewell et al has made critical observations that help to understand the mechanisms of activation of the class IA PI3Ks. By using single-molecule kinetic measurements, the authors have made outstanding progress toward understanding how PI3Kβ is uniquely activated by phosphorylated tyrosine kinase receptors, Gbeta/gamma heterodimers and the small G protein Rac1. While previous studies have defined these as activators of PI3Kβ, the current manuscript makes clear the quantitative limitations of these previous observations. Most previous quantitative in vitro studies of PI3Kβ activation have used soluble peptides derived from bis-phosphorylated receptors to stimulate the enzyme. These soluble peptides stimulate the enzyme, and even stimulate membrane interaction. Although these previous studies showed that the release of p85-mediated autoinhibition unmasks an intrinsic affinity of the enzyme for lipid membranes, they ignored what would be the consequence of these peptide sequences being present in the context of intrinsic membrane proteins. The current manuscript shows that the effect of membrane-conjugated peptides on the enzyme activity is profound, in terms of recruiting the enzyme to membranes. In this context, the authors show that G proteins associated with the membranes have an important contribution to membrane recruitment, but they also have a profound allosteric effect on the activity on the membrane, These are observations that would not have been possible with bulk measurements, and they do not simply recapitulate observations that were made for other class IA PI3Ks.

An important observation that the authors have made is that Gbeta/gamma heterodimers and RAc1 alone have almost no ability to recruit PI3Kβ to the membranes that they are using, and this is central to one of the most profoundly novel activation mechanisms offered by the manuscript. The authors propose that the nSH2- and Gbeta/gamma binding sites partially overlap, so that Gbeta/gamma can only bind once the nSH2 domain releases the p110beta subunit. This mechanism would mean that once the nSH2 is engaged by membrane-congugated pY, the Gbg heterodimer can bind and increase the association of the enzyme with membranes. Indeed, this increased membrane association is observed by the authors. However, the authors also show that this increased recruitment to membranes accounts for relatively little increase in activity, and that the far greater component of activation is due to an allosteric effect of the membrane association on the activity of the enzyme. The proposal for competition between Gbg binding and the nSH2 is consistent with the behavior of an nSH2 mutant that cannot bind to pY and which, consequently, does not vacate the Gbg-binding site. In addition to the outstanding contribution to understanding the kinetics of activation of PI3Kβ, the authors have offered the first structural interpretation for the kinetics of Gbg activation in synergy with pY activation. The proposal for an overlapping nSH2/Gbg binding site is supported by predictions made by John Burke, using alphafold multimer. Although there is no experimental structure to support this structural model, it is consistent with HDX-MS analyses that were published previously.

---

## [Author Response]

The following is the authors’ response to the original reviews.

**eLife assessment**
The manuscript describes the synergy among PI3Kβ activators, providing compelling results concerning the mechanism of their activation. The particular strengths of the work arise to a great extent from the reconstitution system better mimicking the natural environment of the plasma membrane than previous setups have. The study will be a landmark contribution to the signaling field.
**Public Reviews:**

**Reviewer #1 (Public Review):**
The manuscript aims to provide mechanistic insight into the activation of PI3Kβ by its known regulators tyrosine phosphorylated peptides, GTP-loaded Rac1 and G-protein beta-gamma subunits. To achieve this the authors have used supported lipid bilayers, engineered recombinant peptides and proteins (often tagged with fluorophores) and TIRF microscopy to enable bulk (averages of many molecules) and single molecule quantitation. The great strength of this approach is the precision and clarity of mechanistic insight. Although the study does not use "in transfecto" or in vivo models the experiments are performed using "physiologically-based" conditions and provide a powerful insight into core regulatory principles that will be relevant in vivo.The results are beautiful, high quality, well controlled and internally consistent (and with other published work that overlaps on some points) and as a result are compelling. The primary conclusion is that the primary regulator of PI3Kβ are tyrosine phosphorylated peptides (and by inference tyrosine phosphorylated receptors/adaptors) and that the other activators can synergise with that input but have relatively weak impacts on their own.Although the methodology is not easily imported, for reasons of both cost and the experience needed to execute them well, the results have broad importance for the field and reverse an impression that had built in large parts of the broader signalling and PI3K communities that all of the inputs to PI3Kβ were relatively equivalent, however, these conclusions were based on "in cell" or in vivo studies that were very difficult to interpret clearly.
**Reviewer #2 (Public Review):**
The manuscript of Duewell et al has made critical observations that help to understand the mechanisms of activation of the class IA PI3Ks. By using single-molecule kinetic measurements, the authors have made outstanding progress toward understanding how PI3Kβ is uniquely activated by phosphorylated tyrosine kinase receptors, Gbeta/gamma heterodimers and the small G protein Rac1. While previous studies have defined these as activators of PI3Kβ, the current manuscript makes clear the quantitative limitations of these previous observations. Most previous quantitative in vitro studies of PI3Kβ activation have used soluble peptides derived from bis-phosphorylated receptors to stimulate the enzyme. These soluble peptides stimulate the enzyme, and even stimulate membrane interaction. Although these previous studies showed that the release of p85-mediated autoinhibition unmasks an intrinsic affinity of the enzyme for lipid membranes, they ignored what would be the consequence of these peptide sequences being present in the context of intrinsic membrane proteins. The current manuscript shows that the effect of membrane-conjugated peptides on the enzyme activity is profound, in terms of recruiting the enzyme to membranes. In this context, the authors show that G proteins associated with the membranes have an important contribution to membrane recruitment, but they also have a profound allosteric effect on the activity on the membrane, These are observations that would not have been possible with bulk measurements, and they do not simply recapitulate observations that were made for other class IA PI3Ks.An important observation that the authors have made is that Gbeta/gamma heterodimers and RAc1 alone have almost no ability to recruit PI3Kβ to the membranes that they are using, and this is central to one of the most profoundly novel activation mechanisms offered by the manuscript. The authors propose that the nSH2- and Gbeta/gamma binding sites partially overlap, so that Gbeta/gamma can only bind once the nSH2 domain releases the p110beta subunit. This mechanism would mean that once the nSH2 is engaged by membrane-conjugated pY, the Gbg heterodimer can bind and increase the association of the enzyme with membranes. Indeed, this increased membrane association is observed by the authors. However, the authors also show that this increased recruitment to membranes accounts for relatively little increase in activity, and that the far greater component of activation is due to an allosteric effect of the membrane association on the activity of the enzyme. The proposal for competition between Gbg binding and the nSH2 is consistent with the behavior of an nSH2 mutant that cannot bind to pY and which, consequently, does not vacate the Gbg-binding site. In addition to the outstanding contribution to understanding the kinetics of activation of PI3Kβ, the authors have offered the first structural interpretation for the kinetics of Gbg activation in synergy with pY activation. The proposal for an overlapping nSH2/Gbg binding site is supported by predictions made by John Burke, using alphafold multimer. Although there is no experimental structure to support this structural model, it is consistent with HDX-MS analyses that were published previously.
**Reviewer #1 (Recommendations For The Authors):**
1. The approx relative concentrations (surface densities ) of Rac1-GTP, GBetagammas and PY-peptides used in experiments in Fig 1 are not easy to understand and useful to give an intuitive feel for the relative sensitivity of the PI3Kβ reporter to those inputs.

In our revised manuscript, we provide densities of the individual signaling inputs used to reconstitute Dy647-PI3Kβ membrane recruitment (see Figure legend 1). We provide a more detailed explanation about our quantification method in subsequent figures where the membrane surface density of signaling inputs is varied to modulate the strength of PI3Kβ membrane localization and activity.

Building off the quantification of Rac1-GTP and pY membrane density measurements presented in our initial manuscript submission, we now include an estimate of the GβGγ membrane density. For these new measurements, we recombinantly expressed and purified additional SNAP-GβGγ protein, which we fluorescently labeled with AlexaFluor 555. The membrane surface density of GβGγ was quantified at equilibrium using a combination of AF488-SNAP-GβGγ (bulk signal) and dilute AF555-SNAP-GβGγ (0.0025%), which allowed us to resolve and count the single molecule density (Figure 3A). We calculate the total surface density of GβGγ based on the AF555-SNAP-GβGγ dilution factor. In the methods section titled, “surface density calibration,” we describe our protocol.

1. The estimates of the PIP3 concentrations/densities measured using the BTK reporter seem good but its unclear (to me) how they were derived.

The density of PI(3,4,5)P3 lipids in our supported lipid bilayers was calculated based on the incorporation of a define molar ratio of PI(3,4,5)P3 in our small unilamellar vesicles. Based on the average footprint of 0.72 nm2 for a single lipid, we calculated the density of lipids per µm2. In the methods section titled, “kinetic measurements of PI(3,4,5)P3 lipid production,” we include the following description:

“Assuming an average footprint of 0.72 nm2 for phosphatidylcholine (Carnie et al., 1979; Hansen et al., 2019), we calculated a density of 2.8 × 104 PI(3,4,5)P3 lipids/μm2 for supported membranes that contain an initial concentrations of 2% PI(4,5)P2. We assume that the plateau fluorescence intensity of the AF488-SNAP-Btk sensor following reaction completion in the presence of PI3Kβ represents the production of 2% PI(3,4,5)P3. The bulk membrane intensity of AF488-SNAP-Btk was normalized from 0 to 1, and then multiplied times the total density of PI(3,4,5)P3 lipids to generate kinetic traces that report the kinetics of PI(3,4,5)P3 production.”

Minor pointsl164; Rac1(GTP) AND GBeta gammas. In this context it should be OR. Or have I misunderstood?l1093; kineticS measurementS.

Thank you for pointing out these typos. We made the appropriate edits.

The paper of Suire etal (Suire, S., Lécureuil, C., Anderson, K. E., Damoulakis, G., Niewczas, I., Davidson, K., Guillou, H., Pan, D., Jonathan Clark, Phillip T Hawkins, & Stephens, L. (2012). GPCR activation of Ras and PI3Kc in neutrophils depends on PLCb2/b3 and the RasGEF RasGRP4. The EMBO journal, 31(14), 3118-3129. https://doi.org/10.1038/emboj.2012.167) make the point that in vivo it appears that although Ras-activation is required for full activation of PI3Kgamma (and can activate PI3Kgamma in vitro directly) if you use tools to activate Ras in the absence of receptor and Gbetagamma signalling, it has no affect on PIP3 . This directly supports the authors conclusions.

Thank you for sharing this citation. We incorporated the reviewer’s insight into our discussion section to broaden the significance of our work.

**Reviewer #2 (Recommendations For The Authors):**
There are only a few relatively minor points that could be addressed to improve the paper:1. Why is the density still going up after 10 minutes in Figure 1 Figure supplement 2?Doesn't this seem like a very long time? Are we seeing fast on/off combined with fast on/slow off? Are the particles eventually becoming stuck in odd places or are they slowly denaturing?

Our movies do not indicate a slow accumulation of immobilized or stuck Dy647-PI3Kβ particles on the membrane surface. On the long timescale, we believe that a small fraction of Dy647-PI3Kβ molecular do exhibit longer dwell times on membranes containing a high density of pY (>6,000 molecules/µm2). This is likely due to membrane hopping of Dy647-PI3Kβ. In other words, rather than Dy647-PI3Kβ dissociating from the membrane surface directly into the solution, the Dy647-PI3Kβ molecule immediately rebinds to another membrane conjugated pY peptide. This type of behavior of a peripheral membrane binding protein is generally correlated with there being a higher surface density of the binding partner (Yasui et al., 2014). Characterization of potential Dy647-PI3Kβ membrane hopping will require additional experimentation (e.g. PI3Kβ mutants) and quantitative analysis that goes beyond the scope of this study.

1. Lines 188-189. "By quantifying the average number of Alexa488-pY particles per unit area of supported membrane we calculated the absolute density of pY per μm2 (Figure 2D).I think this should be Figure 2C, right hand y-axis.

Thank you for identifying our typo. We’ve corrected the text for clarity.

1. Lines 102-193. "When Dy647-PI3Kβ was flowed over a membrane containing a low density of {less than or equal to} 500 pY/μm2, we observed rapid equilibration kinetics consistent with a 1:1 binding stoichiometry (Figure 2E).” There is no density shown in Fig. 2E. There is only "membrane intensity." Perhaps it was their intent to include a right-hand axis with density (number of particles/area), as they did in Figure 2C. However, they did not, so Figure 2E does not support the text. The value of Intensity/#py/um**2 does not appear to be the same for Figure 2C as for Figure 2E, assuming that the statement in the text is correct. The authors should include the density as a right-hand axis in 2E.

We have reworded this portion of the results section for clarity. In reading the reviewers comment, we recognize that a more convincing way to support our claim of a 1:1 binding stoichiometry would be to show that there are ~500 Dy647-PI3Kβ/μm2 membrane bound complexes when the pY surface density equals ~500 pY/μm2. For us to make this connection, we would need to perform experiments using a Dy647-PI3Kβ concentration that fully saturates all the binding pY binding sites. However, at this elevated Dy647-PI3Kβ solution concentration, individual Dy647-PI3Kβ complexes can start to bind to a single phosphotyrosine of the dually phosphorylated peptide due to competition for pY binding sites. As an alternative to performing the experiment described above, we can infer binding stoichiometry from the shape of the membrane absorption kinetic traces. For example, a simple bimolecular interaction exhibits rapid equilibration kinetics with a hyperbolic shaped kinetic trace. Systems that have more complex binding equilibria, however, generally take longer to equilibrate (due to the change in KOFF) and can often be broken down into 2 or 3 distinct dissociation constants (KD). This type of kinetic analysis has previously been used to describe multivalent membrane binding interactions for the Btk-PI(3,4,5)P3 (Chung et al., 2019) and PI3Kγ-GβGγ (Rathinaswamy et al., 2021) complexes. Considering that there are multiple interpretations of the Dy647-PI3Kβ membrane absorption traces show in Figure 2E, we refrain from saying that our results explicitly reveal a 1:1 binding stoichiometry. Instead, we provide several possible explanations for the results. Ultimately, additional experiments and kinetic modeling of wild type and mutant PI3Kβ is necessary to define the binding stoichiometry under different conditions.

1. Table 1. The authors have analysed the data to extract two dwell times and two diffusion coefficients. The legend should make this clear, referring to D1 as the slow diffusion component and D2 as fast diffusion, similarly, there are short and long dell times. This should be stated in the legend. There are two columns labelled "alpha". This presumably should be alpha1 and alpha2, the fractions of particles with short and long dwell times. The table legend should clarify this.

In our revision, additional text has been added to the figure legends and Table 1.

Text from Table 1: “Alpha (α_τ_) equals the fraction of molecules with the characteristic dwell time, τ1 (DT = dwell time). The fraction of molecules with the characteristic dwell time, τ2, equals 1-α_τ_. Alpha (αD) equals the fraction of molecules with the characteristic diffusion coefficient, D1. The fraction of molecules with diffusion coefficient, D2, equals 1-αD.”

1. In the legend for Figure 5 figure supplement 1, for part D, the "Cumulative membrane of binding events..." The "of" should be deleted.

Thank you for identifying this typo.

1. Lines 423-426: "We found that PI3Kβ kinase activity is also relatively insensitive to eitherRac1(GTP) or GβGγ alone. This is in contrast to previous reports that showed Rho-GTPases (Fritsch et al. 2013) and GβGγ (Katada et al. 1999; Hashem A. Dbouk et al. 2012; Maier, Babich, and Nürnberg 1999) can activate PI3Kβ, albeit modest, compared to synergistic activation with pY peptides plus Rac1(GTP) or GβGγ." It is not clear what this statement means. On the surface, it might be interpreted as saying that these previous studies had some flaw that led the authors to conclude that there is some activation caused by Rac1 or Gbeta/gamma on their own. The current manuscript is an important contribution to understanding the mechanism of synergistic activation, but it is also true that the Hansen and his colleagues have not used the same membranes as were used previously. The authors state that they have used a wide range of membrane compositions, but the only ones that have appeared in the manuscript are nearly pure PC (with 2% PIP2) or PC with 20% PS. Extensive studies with varying membrane compositions are beyond the scope of the current study, since the current manuscript concisely makes important observations regarding mechanism. However, it would be helpful for readers if the authors at least mention the differences in membrane compositions among the studies.

The reviewer raises an important point concerning our interpretation of PI3Kβ activation data in relationship to existing literature. In our original submission, we made conclusions concerning how individual signaling inputs modulate PI3Kβ activity, without showing all our data or providing sufficient explanation. In our revised manuscript, we include PI3Kβ kinase activity measurements performed in the presence of either pY, Rac1(GTP), or GβGγ alone (Figure 5B-5C). These experiments were reconstituted on supported membranes in the absence or presence of 20% PS lipids. We found that increasing the density of anionic lipids increased the overall activity of PI3Kβ in the presence of pY or GβGγ alone. This is consistent with a subtle increase in PI3Kβ membrane affinity due to the negatively charged PS lipids. Mutations that disrupt the direct interaction between PI3Kβ and GβGγ eliminated the observed lipid kinase activity. We were unable to detect PI3Kβ activity in the presence of Rac1(GTP) alone. In conclusion, we’re able to detect some PI3Kβ activity in the presence of GβGγ alone, which is consistent with previous reports (Dbouk et al., 2010; Katada et al., 1999; Maier et al., 2000). In the future, a more comprehensive analysis will be required to map the relationship between PI3Kβ activity, membrane localization, and lipid composition. For example, previous reconstitutions have revealed differential activation of PI3Kα that depends on the most abundant lipid being phosphatidylethanolamine (PE) rather than phosphatidylcholine (PC) (Hon et al., 2012; Ziemba et al., 2016). PE lipids comprise 25-30% of the cellular plasma membrane (Yang et al., 2018) and have been used in previous studies to measure PI3K lipid kinase activity on small unilamellar vesicles (Dbouk et al., 2010; Hon et al., 2012).

In this study, we elected to use a simplified membrane composition that minimized non-specific membrane localization of fluorescently labeled PI3Kβ. This allowed us to more clearly define the strength of individual and combinations of protein-protein interactions that regulate PI3Kβ localization and kinase activity. When reconstituting amphiphilic molecules (i.e. lipids) in aqueous solution a variety of structures, including micelles, inverted micelles, and planar bilayers can form based on the lipid composition (Kulkarni, 2019). The organization of these membrane structures is related to the molecular packing parameter of the individual phospholipids (Israelachvili et al., 1976). The packing parameter (P=v⁄((a•l_c))) depends on the volume of the hydrocarbon (v), area of the lipid head group (a), and the lipid tail length (l_c). When generating supported lipid bilayers on a flat two-dimensional glass surface, we aim to create a fluid lamellar membrane. We find that phosphatidylcholine (PC) lipids are ideal for making supported lipid bilayers because they have a packing parameter of ~1 (Costigan et al., 2000). In other words, PC lipids are cylindrical like a paper towel roll. In contrast, cholesterol and phosphatidylethanolamine (PE) lipids have packing parameters of 1.22 and 1.11, respectively (Angelov et al., 1999; Carnie et al., 1979). This gives cholesterol and PE lipids an inverted truncated cone shape, which prefers to adopt a non-lamellar phase structure. Due to the intrinsic negative curvature of PE lipids, they can spontaneously form inverted micelles (i.e. hexagonal II phase) in aqueous solution when they are the predominant lipid species (Israelachvili et al., 1980; Kobierski et al., 2022; Wnętrzak et al., 2013). In the methods section of our manuscript, we note that from our experience incorporation of PE lipids dramatically reduces the protein-maleimide coupling efficiency, displayed more membrane defects, and resulted in a larger fraction of surface immobilized Dy647-PI3Kβ. This could be related to the intrinsic negative curvature of PE membranes. However, further investigation is needed to decipher these issues.

Angelov B, Ollivon M, Angelova A. 1999. X-ray Diffraction Study of the Effect of the Detergent Octyl Glucoside on the Structure of Lamellar and Nonlamellar Lipid/Water Phases of Use for Membrane Protein Reconstitution. Langmuir 15:8225–8234. doi:10.1021/la9902338

Carnie S, Israelachvili JN, Pailthorpe BA. 1979. Lipid packing and transbilayer asymmetries of mixed lipid vesicles. Biochim Biophys Acta 554:340–357. doi:10.1016/0005-2736(79)90375-4

Chung JK, Nocka LM, Decker A, Wang Q, Kadlecek TA, Weiss A, Kuriyan J, Groves JT. 2019. Switch-like activation of Bruton’s tyrosine kinase by membrane-mediated dimerization. Proc Natl Acad Sci 116:10798–10803. doi:10.1073/pnas.1819309116

Costigan SC, Booth PJ, Templer RH. 2000. Estimations of lipid bilayer geometry in fluid lamellar phases. Biochim Biophys Acta 1468:41–54. doi:10.1016/s0005-2736(00)00220-0

Dbouk HA, Pang H, Fiser A, Backer JM. 2010. A biochemical mechanism for the oncogenic potential of the p110 catalytic subunit of phosphoinositide 3-kinase. Proc Natl Acad Sci 107:19897–19902. doi:10.1073/pnas.1008739107

Hansen SD, Huang WYC, Lee YK, Bieling P, Christensen SM, Groves JT. 2019. Stochastic geometry sensing and polarization in a lipid kinase–phosphatase competitive reaction. Proc Natl Acad Sci 116:15013–15022. doi:10.1073/pnas.1901744116

Hon W-C, Berndt A, Williams RL. 2012. Regulation of lipid binding underlies the activation mechanism of class IA PI3-kinases. Oncogene 31:3655–3666. doi:10.1038/onc.2011.532

Israelachvili JN, Marcelja S, Horn RG. 1980. Physical principles of membrane organization. Q Rev Biophys 13:121–200. doi:10.1017/s0033583500001645

Israelachvili JN, Mitchell DJ, Ninham BW. 1976. Theory of self-assembly of hydrocarbon amphiphiles into micelles and bilayers. J Chem Soc Faraday Trans 2 Mol Chem Phys 72:1525–1568. doi:10.1039/F29767201525

Katada T, Kurosu H, Okada T, Suzuki T, Tsujimoto N, Takasuga S, Kontani K, Hazeki O, Ui M. 1999. Synergistic activation of a family of phosphoinositide 3-kinase via G-protein coupled and tyrosine kinase-related receptors. Chem Phys Lipids 98:79–86. doi:10.1016/S0009-3084(99)00020-1

Kobierski J, Wnętrzak A, Chachaj-Brekiesz A, Dynarowicz-Latka P. 2022. Predicting the packing parameter for lipids in monolayers with the use of molecular dynamics. Colloids Surf B Biointerfaces 211:112298. doi:10.1016/j.colsurfb.2021.112298

Kulkarni CV. 2019. Calculating the “chain splay” of amphiphilic molecules: Towards quantifying the molecular shapes. Chem Phys Lipids 218:16–21. doi:10.1016/j.chemphyslip.2018.11.004

Maier U, Babich A, Macrez N, Leopoldt D, Gierschik P, Illenberger D, Nürnberg B. 2000. Gβ 5 γ 2 Is a Highly Selective Activator of Phospholipid-dependent Enzymes. J Biol Chem 275:13746–13754. doi:10.1074/jbc.275.18.13746

Rathinaswamy MK, Dalwadi U, Fleming KD, Adams C, Stariha JTB, Pardon E, Baek M, Vadas O, DiMaio F, Steyaert J, Hansen SD, Yip CK, Burke JE. 2021. Structure of the phosphoinositide 3-kinase (PI3K) p110γ-p101 complex reveals molecular mechanism of GPCR activation. Sci Adv 7:eabj4282. doi:10.1126/sciadv.abj4282

Wnętrzak A, Lątka K, Dynarowicz-Łątka P. 2013. Interactions of alkylphosphocholines with model membranes-the Langmuir monolayer study. J Membr Biol 246:453–466. doi:10.1007/s00232-013-9557-4

Yang Y, Lee M, Fairn GD. 2018. Phospholipid subcellular localization and dynamics. J Biol Chem 293:6230–6240. doi:10.1074/jbc.R117.000582

Yasui M, Matsuoka S, Ueda M. 2014. PTEN Hopping on the Cell Membrane Is Regulated via a Positively-Charged C2 Domain. PLoS Comput Biol 10:e1003817. doi:10.1371/journal.pcbi.1003817

Ziemba BP, Burke JE, Masson G, Williams RL, Falke JJ. 2016. Regulation of PI3K by PKC and MARCKS: Single-Molecule Analysis of a Reconstituted Signaling Pathway. Biophys J 110:1811–1825. doi:10.1016/j.bpj.2016.03.001